

**Assessing raindrop evolution over northern Western Ghat**
**from stable isotope signature of rain and vapour**
Sheena Sunil Nimya [1,2], Sundara Pandian Rajaveni[1], Saikat Sengupta[1*], Sourendra Kumar
Bhattacharya[3]
[1]Center for Climate Change Research, Indian Institute of Tropical Meteorology, Ministry of Earth Sciences,
Pune-411008, India
[2]Department of Earth, Atmospheric and Planetary Sciences, Purdue University, West Lafayette, IN, USA
[3] Institute of Earth Sciences, Academia Sinica, Taipei 11529, Taiwan
*Correspondence to: Saikat Sengupta, email: saikat@tropmet.res.in
**Abstract.** Isotope exchange between vapor and rain critically influences rain isotope values, which are useful in
modeling raindrop evolution. A one-dimensional Below Cloud Interaction Model (BCIM) has been used to
quantify sub-cloud processes affecting raindrop evolution in extratropical regions. However, its applicability has
not been tested in a tropical monsoon region, where both advection of moisture and raindrop evaporation are
significant. Here, we evaluate the applicability of BCIM using simultaneous surface measurements of rain and
vapor isotopes over Pune, a tropical rain-shadow region, during the 2019 Indian Summer Monsoon. Analysis of
these data indicates strong isotope exchange and significant raindrop evaporation in the sub-cloud layer. A
Rayleigh ascent in BCIM overestimates rain isotope values (by about 6 ‰ for $\delta$D), although model and
observed values are well correlated. Using radiosonde-based temperature and humidity profiles and constructing
vapour isotope profiles from a combination of satellite (Tropospheric Emission Spectrometer) data and the
LMDZ model outputs, simulations improve. Further tuning of vapour isotope inputs while preserving the shape
of the profiles yields still better agreement. Sensitivity studies reveal that model outputs are strongly influenced
by vapour isotope profiles, and moderately by drop size and relative humidity. We used BCIM to estimate
raindrop evaporation, which shows that, on average, 23 % of rain mass evaporated over Pune. Our results
emphasize the importance of rain evaporation over the Indian continent during the Monsoon season, in
particular, over complex orography, and illustrate the use of water isotopes to constrain this key process.



## 1. Introduction

The Intergovernmental Panel on Climate Change (IPCC) has emphasised the importance of recycled moisture in the atmosphere (IPCC, 2014). Moisture recycling includes processes by which a fraction of the precipitated water returns to the atmosphere and adds to the existing vapour that may cause further precipitation over the same area (Gray, 2012). These processes are soil evaporation, transpiration from plants, intercepted or condensed water on leaves, and evaporation from falling raindrops (Brubaker et al., 1993; Trenberth, 1999). The moisture recycling increases with the ambient temperature but is lessened with increasing humidity (Pranindita et al., 2022; Zaitchik et al., 2006; Zhang et al., 2021). Some earlier studies have estimated that a high precipitation recycling ratio, the ratio of recycled precipitation to total precipitation, operates over India (on average 15 %) during the Indian Summer Monsoon (ISM; June-September); this happens despite the high humidity that prevails over the subcontinent (Kumar et al., 2021; Pathak et al., 2014). Among the recycled moisture sources, raindrop evaporation is difficult to estimate because (1) determination of the parameters needed for estimating rain evaporation from various satellite data is not sufficiently accurate, and (2) station-based meteorological observations using Micro rain radars are limited (Dai et al., 2019; Li and Srivastava, 2001; Xie et al., 2016).

Stable isotopologues (mainly $^1H_2^{18}O$, $^1H^2H^{16}O$, $^1H_2^{16}O$) of liquid and solid precipitation samples can be used to assess the magnitude of raindrop evaporation (Crawford et al., 2017; Rahul et al., 2016; Salamalikis et al., 2016; Wang et al., 2021; Xiao et al., 2021). Falling raindrops exchange isotopes with the ambient vapour; this happens throughout the fall but occurs mostly in the unsaturated sub-cloud layer. The magnitude of this exchange, which alters the rain isotope ratios, can, in principle, be used to quantify the extent of raindrop evaporation. Using satellite-based observations of vapour isotopologues ($^1H^2H^{16}O$ and $^1H_2^{16}O$) and an isotope mass balance model, Worden et al. (2007) estimated that in the tropics, during the October to March interval, nearly 20 % of the mass of a raindrop evaporates. However, they also mentioned that the satellite data used for this Estimate has limited temporal and spatial coverage. Therefore, these datasets may not be useful for estimating drop evaporation on a daily to monthly scale over some specific locations. In another approach, raindrop evaporation has been estimated from ground-based rain isotope observations and a set of empirical equations (Froehlich et al., 2008; Li et al., 2021; Wang et al., 2016; Zhu et al., 2021). However, such attempts are often inaccurate because they exclude many important cloud microphysical processes and associated isotopic fractionations. Normally, these processes are considered for simulating rain isotope values in various General Circulation Models (GCM; Risi et al., 2019; Yoshimura et al., 2008). The GCMs incorporate the isotope exchange scheme associated with evaporation (Stewart, 1975). Nevertheless, recent studies have shown that most of these GCMs over or underestimate raindrop evaporation in tropical India (Nimya et al., 2022; Sengupta et al., 2023). This is possibly due to the coarseness of grid sizes used in these GCMs, which are inadequate to capture the region-specific complexities of processes controlling the evaporation. This necessitates controlled isotope observations and region-specific models for a reasonable estimation of this parameter (Aemisegger et al., 2015).

Various modelling approaches were followed to estimate raindrop evaporation using paired observations of rain and vapour isotopes. For example, a bin resolved microphysical model was used to quantify drop evaporation during the Atlantic Tradewind Ocean–Atmosphere Mesoscale Interaction Campaign



(ATOMIC; Sarkar et al., 2023). Graf et al. (2019), based on surface rain and vapour isotope observations in
Zurich, Switzerland, provided a rationale to evaluate various processes controlling the isotope values. They
developed a simple one-dimensional model (Below Cloud Interaction Model, BCIM) which considers essential
cloud microphysical processes during raindrop formation (vapour deposition, rimming etc.) as well as
evaporative exchange processes below the cloud. That model, in principle, showcases the isotopic evolution of
an ice/liquid drop that is released from a desired altitude and suffers the aforementioned processes enroute its
fall to the ground. Although their model is capable of differentiating isotopic signals of different sub-cloud
processes, it does not consider any moisture advection, updraft and downdraft. It is worthwhile to explore the
efficacy of that model in a semi-tropical region during ISM when advected moisture fluxes play an important
role (Das, 1986; Levine and Turner, 2012).

In the Western Ghat (WG) region, shallow convective (80 % of clouds occur below 4 km and 45 %

below 2.5 km altitude) clouds predominate during the ISM (Konwar et al., 2014). Faster evaporation of smaller
raindrops associated with intense rainfalls from these clouds provides significant positive energy feedback to
form mesoscale convection (Konwar et al., 2014; Tao et al., 2012). Another study, based on drop size
distributions, showed that raindrop evaporation prevails in the warm rain process (shallow clouds) in this region
(Murali Krishna et al., 2021). However, these studies were limited to scanty observations. The question arises of
whether one can determine the raindrop evaporation and its variation using an independent, accurate, and
simpler method. Isotope ratios in rain and vapour provide such a method.

The current study investigates the applicability of the BCIM in a tropical Indian rain shadow region

using paired observations of rain and vapour isotopes for a summer monsoon season. By a suitable choice of
input parameters for the BCIM, we can estimate the raindrop evaporation in this tropical zone.

**2. Experimental Methodology**
**2.1 Study area**

Rainwater and vapour samples (mostly on rainy days but even for some non-rainy days) were collected from the
near-surface premise of the Indian Institute of Tropical Meteorology (18.53° N, 73.85° E), Pune during the
summer monsoon of 2019. This region receives >90 % rainfall during the ISM and is situated at the lee (rain
shadow) side of the Mountain (Fig. 1). A brief discussion on sources and mechanisms of the ISM rainfall in
western India is given below. Rainfall in Western India occurs from mid-tropospheric low-pressure systems in
episodes, each of which usually lasts for 2–3 days; these systems are locked in place during these periods and
fed by moisture derived from the Arabian Sea (Wang et al., 2006; Rao, 1976). The geographic location of the
region, its altitude (from mean sea level), rainfall variation across the WG mountains, and the altitude
topographic profile across Pune are shown in Fig. 1.

There is a sharp variation of rainfall across the mountain from the coastal zone (30 mm day$^{-1}$) to the lee

side (12 mm day$^{-1}$) which is a characteristic of orography-induced rainfall (Fig. 1). The surface air temperature
varies from 23.1° C to 18.4° C during the ISM (Pattanaik et al., 2019).






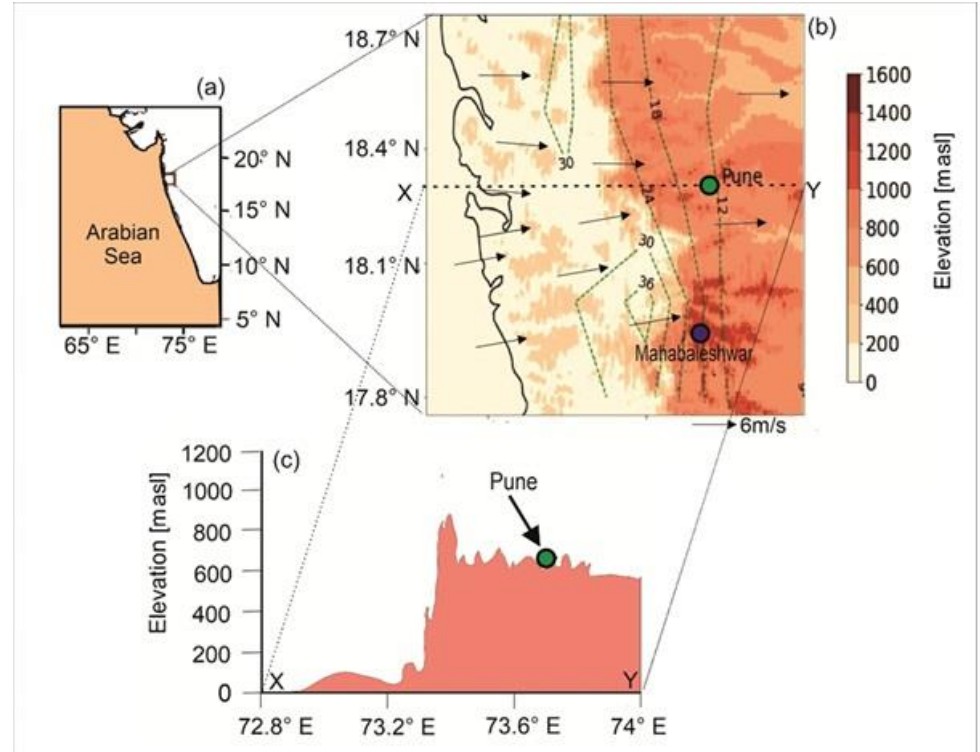


**Figure 1: (a)** The location of the study area in India. **(b)** Topographic map of the northern Western Ghat, India (prepared based on the GTOPO30 digital elevation model). The rainfall contours (long-term (1901-2017) mean June-September rainfall in mm/day) were constructed using gridded (0.25°x0.25°) rainfall data (1901-2020) from the India Meteorological Department(IMD). **(c)** A topographic profile along the latitude 18.53° N through Pune (Green circle) shows its position in a rain shadow region.

**2.2 Sample Collection, Isotope Measurements**

The onset and withdrawal dates of ISM (based on the wind direction, specific humidity, and outgoing long wave radiation (OLR; IMD, 2019) at Pune in 2019 were 22 June 2019 and 4 October 2019, respectively. Liquid water samples were collected during rains using samplers made following the guidelines of the International Atomic Energy Agency (http://www.naweb.iaea.org/napc/ih/documents/other/gnip_manual_v2.02_en_hq.pdf). The samples were transferred into 8 ml Polycarbonate bottles and stored in a dark and cold place. The sampling procedure and storage were such as to ensure negligible evaporation of the samples after collection (Rajaveni et al., 2024). A total of 59 rain samples were collected. An indigenously fabricated glass condenser was used for the vapour sample collection. One end of the condenser was connected to a diaphragm pump through a PTFE tube. The ambient air was pulled by the pump through this glass condenser at a flow rate of 800 ml min-1. The glass condenser was immersed in a precooled Dewar flask. The temperature of the flask was maintained at -80° C by using alcohol slurry. At this temperature, the atmospheric moisture is condensed into water, and all other





non-condensable gases are pumped out. The sampling was usually carried out for 3-4 hours, which was
sufficient to obtain an adequate amount of water (the collected sample was at least double the minimum amount
required for isotope analysis, ~1 ml). Most of the samples were collected during the rainy days (avoiding direct
raindrop entry), but some were collected during the non-rainy period. The collected samples are transferred to 2
ml glass vials, which are directly used for isotopic measurement. Due to logistical problems, vapour samples
could not be collected before mid-July. A total of 50 vapour samples were collected during the study period, and
29 of them coincided with rainy days. The liquid samples (both water and condensed vapour) were measured
using a Liquid Water Isotope Analyser (Model Number TIWA-45-EP) manufactured by Los Gatos Research
(LGR). This instrument measures liquid samples using Off-Axis integrated cavity output spectroscopy (OA-
ICOS) with a routine precision of 0.1 ‰ and 1 ‰ for $\delta^{18}O$ and $\delta D$, respectively (Rajaveni et al., 2024). The d-
excess values defined as: d-excess = $\delta D - 8 * \delta^{18}O$ (Dansgaard, 2012) have a precision of 1 ‰. The daily rain
isotope data are weighted by the amount of rainfall on that day.

**2.3 Satellite and ground-based meteorological observations**

The rainfall data (cumulated over 24 hours) are obtained from the Pune observatories of the IMD (available at
the National Data Centre (www.imdpune.gov.in/ndc_new/ndc_index.html)). Apart from daily rainfall, hourly
rainfall, daily average temperature and relative humidity data for Pune observatory were also obtained from the
IMD. The daily gridded data (zonal and meridional wind, specific humidity, air temperature, and cloud liquid
water content) from the European Centre for Medium-Range Weather Forecasts Reanalysis (ERA-5) dataset
with a resolution of 0.25° × 0.25° (Hersbach et al., 2020) are also used. The Interpolated Outgoing Longwave
Radiation (OLR) data (2.5° × 2.5°) from NOAA (https://psl.noaa.gov/data/gridded/data.olrcdr.interp.html) are
used in this study. The upper-air radiosonde measurements (relative humidity, temperature) were obtained from
the University of Wyoming repository (http://weather.uwyo.edu/upperair/sounding.html) in February 2023. The
radiosonde data were available over Pune at 00 UTC and 12 UTC for the entire study period. The two profiles
are averaged to make a representative daily profile for the study period. The typical uncertainty of temperature
and relative humidity is 0.5° C (Jensen et al., 2016) and 5 % (Xu et al., 2023), respectively. Tropospheric
Emission Spectrometer (TES) Level 2 (Nadir-Lite-Version 6) retrievals of HDO and $H_2O$ profiles for the
available period (2005–2007) are used to construct a mean vapour $\delta D$ profile. The details of quality control
criteria and biases associated with TES observations are discussed by Herman et al. (2014) and Worden et al.
(2011). Grid point observations of $\delta D$ by TES have a precision of ~ 10–15 ‰, which reduces to 1–2 ‰ when
the data are averaged over a larger region (Lee et al., 2011; Pradhan et al., 2019).
To decipher the moisture sources for vapour/rain at and around our study area, 48 h air mass back
trajectory analysis was carried out at 850 mb pressure level using the NOAA Hybrid Single-Particle Lagrangian
Integrated Trajectory (HYSPLIT) model (Draxler and Hess, 1997). The model tracks the movement of air
parcels backward from a given location for a desired period. The Global Data Assimilation System (GDAS; 1° ×
1°; Kanamitsu, 1989) dataset is used for back-trajectory analyses.


**2.4 Isotope Model BCIM**




As mentioned before, to understand water vapour isotope exchange in the sub-cloud layer, we used the Below
Cloud Interaction Model (BCIM) proposed by Graf et al. (2019). Various parameterisation schemes used in the
BCIM have been discussed in the aforementioned earlier study. A brief description of this model, as applicable
for Pune (shallow cloud processes), is provided for completeness. The model comprises a single vertical column
that extends from the ground level to the point at which a single hydrometeor is introduced at the base of the
cloud, and follows its fate. Within this column, the hydrometeor descends under the influence of gravity,
undergoes growth or evaporation (depending upon the ambient humidity and temperature), changes its isotopic
composition through equilibrium and kinetic isotope exchange with surrounding vapour, and finally reaches the
surface as rain. The final isotopic composition of the hydrometeor is estimated following four steps of
calculations: (1) setting up the initial condition, (2) estimation of the initial isotopic composition of the
hydrometeor, (3) micro-physics of falling hydrometeor, and (4) tracking the changes in isotopic composition
along the vertical fall trajectory. To estimate the initial isotopic composition of the drop and its evolution, the
model requires temperature, humidity and vapour isotope depth profiles for a given day as input parameters. The
drop is assumed to form in equilibrium (at relative humidity, RH=100 %) at the cloud base and starts its
journey. The input parameters applicable to the vapour can be introduced into the model in two different ways:
(1) the profiles can be calculated based on the idealised (moist) adiabatic ascent of an air parcel from the surface
to the top of the column following a Rayleigh model; isotope values at various pressure levels are then estimated
from the Rayleigh distillation equation, and (2) the pressure level specific values of the aforementioned
parameters, if available from radiosondes and any model, can be introduced directly into the BCIM.
Apart from temperature, humidity and vapour isotopes, the model also requires a drop diameter of the
initial hydrometeor (given in Section 4.3.1.1). Next, the isotopic composition of the introduced hydrometeors is
estimated. They are assumed to be formed in equilibrium from the vapour at this altitude, and their composition
is calculated from the isotopic composition of this vapour at ambient temperature. Subsequently, these drops
grow or diminish as they fall. The isotopic composition of the falling hydrometeor at a given altitude is then
estimated from the composition of the surrounding vapour by using isotope mass balance and diffusive transport
involving appropriate fractionation factors (Graf et al., 2019).
The mass and temperature of the hydrometeor are calculated along its fall trajectory through the
microphysics of the falling hydrometeor. The terminal velocities are estimated using Foote and du Toit (1969).
To calculate the change in mass and temperature between two pressure levels, the temperature, pressure, and
humidity values are interpolated between the two levels. These changes are estimated as per Pruppacher and
Klett (2010). It is important to mention here that many processes considered in BCIM do not occur for the
shallow convective clouds in Pune (Utsav et al., 2017). Therefore, the BCIM as given in Graf et al. (2019) is
modified in the present study.

**3. Results**

Measured rain and vapour isotope ratios ($\delta^{18}$O and d-excess) on a daily scale are plotted in Fig. 2a and 2b. The
general pattern of variations in vapour and rain $\delta^{18}$O values is similar; both decrease significantly and
consistently after mid-August. The vapour $\delta$-values are lower than the rain. In contrast, the d-excess values of





vapour are always much higher. The $\delta^{18}$O and d-excess values of rainwater range from −10.8 ‰ to 1.5 ‰ and
−2 ‰ to 12 ‰, while those of the vapour range from −19 ‰ to −9 ‰ and 10 ‰ to 30 ‰, respectively. The
mean and 0.5σ standard deviation of $\delta^{18}$O and d-excess values of rainwater are −1.3±1.2 ‰ and 3.9±1.3 ‰,
while those of the vapour are -12.5±1.25 ‰ and 18.3 ± 2.55 ‰, respectively. The $\delta^{18}$O (Fig. 2a) and d-excess
(Fig. 2b) time series show four interesting features: (1) For the four date ranges: 27-29 July, 24-27 July, 4-8
September, 19-27 September, significant and consistent decrease in isotope values are observed in both rain and
vapour phases (marked 1, 2, 3, 4 in Fig. 2a), (2) On 19 September, the vapour shows sudden decrease (marked
A in Fig. 2a), (3) Gradual decrease in vapour $\delta^{18}$O values and increase in d-excess values are observed with
progress of monsoon, especially more in the later part, and (4) Rain d-excess values remained essentially
constant with time but $\delta^{18}$O of both rain and vapour started decreasing beginning from early September
onwards.

The rain and vapour isotopic depletion in the tropics is often associated with mesoscale convection

(Lekshmy et al., 2014; Risi et al., 2008; Sengupta et al., 2020). We define depleted-isotope events as those
where isotope ratios of a group of samples fall below the overall mean (μ)-0.5 standard deviation (σ; Sengupta
et al., 2020). To examine the extent to which the depleted (more negative) isotope events are related to large
convective events, a latitude-time Hovmoeller plot of daily OLR anomaly (averaged over the longitude 70° E -
75° E) is displayed in Fig. 2c. The OLR values are often used as a proxy for convection in tropical and
subtropical regions. Since cloud top temperatures (colder is higher) are an indicator of cloud height, negative
OLR anomaly means colder cloud top temperatures or higher cloud thickness. This, in turn, implies extensive
coverage by deep cloud systems characteristic of mesoscale convection and rain. A time synchronous
association of low OLR and low isotope events thus indicates mesoscale convection affecting isotope values.
Fig. 2c indicates four such isotope-depleting mesoscale events (marked as 1, 2, 3 and 4). In addition, we also see
one depleted isotope event without such association (marked as A in the Fig. 2c). We note from Fig. 2d that
major rainfall occurred during the months of July and August; the relative humidity at the surface during the
whole period varied from 57 % to 99 %, and the surface temperature varied from 22° C to 32° C (not shown). It
is evident from the figure that deep convection is associated with high rainfall for three events (1, 2, and 4). A
recent study, based on a year-long continuous measurement of atmospheric vapour, also noticed such isotopic
depletion during high rainfall events over a northern tropical station in Sri Lanka (Wu et al., 2025).

As mentioned, an increasing trend (13 ‰ to 30 ‰) in the vapour d-excess values associated with a

decrease in the $\delta^{18}$O values is noted with the progress of the monsoon (Fig. 2b). In contrast, the rain d-excess
values were reasonably constant within a small range. The increase in vapour d-excess (and decrease in the
$\delta^{18}$O) is large and could be ascribed to significant recycling of the moisture with contribution from some
evaporative sources (discussed later). We are not certain about the source at this stage. Risi et al. (2023) have
discussed the possibility of down-drafted vapours as the source of such anomalously low isotope ratios in the
case of Sahelian squall lines. Earlier studies over some Indian sites have shown that changes in moisture sources
are often associated with a concomitant change in isotope values in rain and vapours (Deshpande et al., 2010;
Midhun et al., 2018). We investigated the possibility of this by forty-eight hours of air-parcel back trajectory
analysis (Supplementary Fig. S1), which shows that moisture for the 2019 summer monsoon season was derived
mainly from the Arabian Sea. However, this does not rule out the possibility of minor contributions from



continental moisture sources or down-drafted moisture characterised by low isotope ratios and high d-excess values (Risi et al., 2010).

**Figure 2.** The time series of $\delta^{18}O$ **(a)** and d excess values, **(b)** of the rainwater (RW) and water vapour (WV), **(c)** OLR anomaly (W m$^{-2}$), and **(d)** daily rainfall (mm over 24 h; d) in Pune. The four shaded vertical bars (numbered 1, 2, 3, and 4) denote synchronous low OLR values and low isotope values (i.e., less than their respective µ-0.5σ values). These periods are defined as low-isotope events. A indicates one isolated low isotope value without low OLR. Thick arrows show how convective cloud bands (indicated by low OLR anomaly) traverse to the sampling region over Pune.



Fig. 3a shows the local meteoric water line (LMWL) using rainwater samples and the local water

vapour line (LWVL) using vapour samples, both pertaining to the monsoon period. The LMWL equation is $\delta D_r$

= (7.3±0.1) $\delta^{18}O$ + (3.0±0.3) and the LWVL, $\delta Dv$ = (6.4±0.2 $\delta^{18}O$) – (1.9±3.0). The slope and intercept of the

LMWL values are lower than those of the Global Meteoric Water Line (GMWL), which are 8.0 and 10.0,

respectively (Dansgaard, 2012; Gat, 1996). This difference, though small, suggests some amount of below-cloud

evaporation of the rains. At Roorkee, a high-latitude Indian Station, Saranya et al. (2018) found an LMWL with

a lower slope (5.4) but a higher intercept (27) compared to our Pune values. They attributed these changes to the

contribution of evaporation from water bodies nearby and moisture recycling during the monsoon. Rahul et al.

(2016) got a similar slope (7.4) but a lower intercept (1.5) in Bangalore (southern central India, at a high altitude

of ~1 km). The slopes of meteoric water lines provide a signature of evaporation processes associated with

kinetic fractionations occurring during rainfall events.

The d-excess values of rain samples suffering evaporation generally bear a negative relationship with

$\delta^{18}O$ values (Bonne et al., 2014; Munksgaard et al., 2020). This is seen in our study (Fig. 3b) where rain d-

excess increases with a decrease in $\delta^{18}O$ values. In addition, the vapour d-excess values also show a statistically

significant negative correlation with $\delta^{18}O$ values (Fig. 3b; $R^2$ = 0.61; p = 0.001), probably indicating

contribution of vapour derived from rain evaporation (Kurita, 2013; Risi et al., 2021). Correlation studies can be

indicative, but the causative factors behind the above variations can be explored with the help of a process-based

model. Below, the role of local meteorological factors and rain-vapour isotope exchange will be explored with

the help of BCIM.

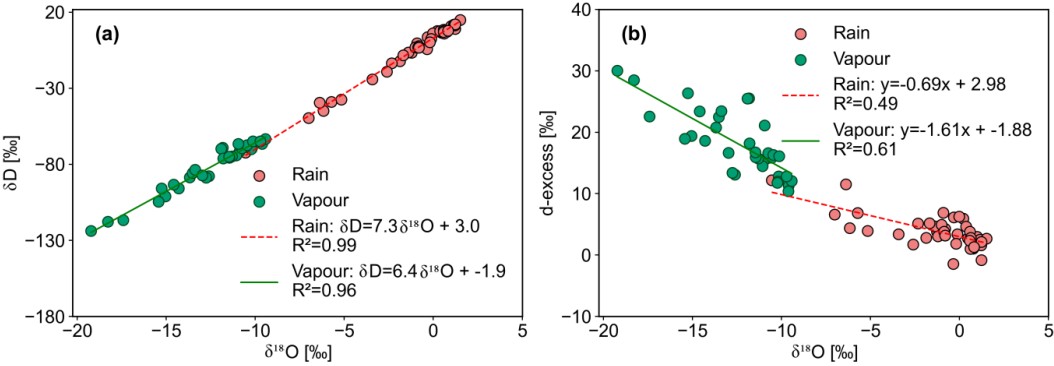



**Figure 3.** A cross-plot of **(a)** $\delta D$ and $\delta^{18}O$ of rain and vapour; **(b)** a cross-plot of d-excess and $\delta^{18}O$ of rain and vapour. Mean
regression lines and correlation coefficients are shown inside the plots.

**4. Discussion**
**4.1 Influence of local meteorological parameters on isotopes**

Water isotopes in the tropics often vary with rainfall, humidity, and temperature (Dansgaard, 2012; Lee and
Fung, 2008). Scatter plots between the vapour d-excess values and local meteorological parameters such as
rainfall amount, relative humidity, specific humidity and temperature are shown in Supplementary Fig. S2. The
d-excess of vapour shows only a marginal positive correlation with temperature ($R^2$=0.16; p-value=0.03; not



significant) and a small negative correlation with relative humidity ($R^2=0.22$; p-value=0.01; marginally
significant).

It is known that temperature and relative humidity of air have opposite controls on raindrop

evaporation (Lee and Fung, 2008; Stewart, 1975). No significant correlations (not shown) are found between the
rainwater isotopes and rainfall. This is contrary to the anti-correlation found in other climate zones (Lee and
Fung, 2008). The absence of correlation in tropics is also found in many recent studies (Chakraborty et al.,
2016; Moerman et al., 2013; Vimeux et al., 2011). In fact, a correlation is often found with the regional
convective activities (Kurita, 2013; Lekshmy et al., 2018). Risi et al. (2023) have noted that in the tropics, most
of the precipitation falls under deep convective systems, which are controlled by different microphysical
processes (like rain evaporation, diffusive liquid-vapour exchanges, and mesoscale downdrafts) connected
through mesoscale circulations.

**4.2 Rain-vapour isotope exchange and rain evaporation**

The micro-physical process of evaporative exchange during the fall of raindrops causes isotopic enrichment in
the rain. Though important, raindrop evaporation cannot be easily quantified. As discussed before, evaporation
is reflected in the higher δ-values and lower d-excess values (mean~2 ‰) of the rain samples. Froehlich et al.
(2008) used d-excess values of precipitation in the Alpine region to derive the extent of evaporation using
assumed end-member values of the regional vapours.

To inspect the isotope exchange between the rain and ambient vapour, the isotope data for the dates

when both rain and vapour samples were collected are analysed here. A strong correlation between rain and
vapour $\delta^{18}O$ values is found (Fig. 4a; $R^2=0.64$, p < 0.01, n=29), suggesting a genetic connection between them.
Sinha and Chakraborty (2020) also found significant positive relations ($R^2>0.8$) between rain and vapour $\delta^{18}O$
values over Andaman Island. However, they did not find any anti-correlation between rain $\delta^{18}O$ and rain d-
excess, as we did (Fig. 3b). The current study exhibits a reasonable anti-correlation between the differences in d-
excess (Δd-excess (r-v)) and $\delta^{18}O$ ($\Delta\delta^{18}O(r-v)$) of rain and vapour (Fig. 4b). This would be expected if
evaporation of rain contributes a significant amount of vapour because the inherited vapour is lower in $\delta^{18}O$ but
higher in d-excess compared to the rain.

As raindrops evaporate, the newly formed vapour may get down-drafted to the low level vapour, and

therefore, the two phases at the ground would exhibit opposite changes. Interestingly, in the case of tropical
precipitation, we do not expect a substantial contribution from rain evaporation to the background vapour
because the latter is a large reservoir. It has been shown in several earlier studies that the total rain is derived
from only a few percent of the overhead vapour mass (Pathak et al., 2014; Rahul et al., 2016). Earlier studies
have also shown that vapour d-excess values do not exhibit any systematic change in central or southern WG
stations, although, surprisingly, their rain $\delta^{18}O$ values exhibit slight but gradual depletion (1 ‰ to -10 ‰) in the
later part of the monsoon (Lekshmy et al., 2018; Rahul et al., 2016). The negative correlation found in this study
suggests that the ground-level vapour gets a significant contribution from drop evaporation. How can moisture
generated by drop evaporation over the falling path contribute to the ground-level vapour?. This is possible
when there is a strong downdraft associated with intense monsoon rains (Risi et al., 2023). In a modelling study,
Mandke et al. (1999) pointed out that deep convective cloud systems contain both upward and downward





components. The downward motion is driven by the evaporation of falling precipitation and the dragging of the
ambient air and vapour by big droplets. This downdraft brings moisture down from above and increases the
vapour d-excess at the surface (Risi et al., 2010; Kurita, 2013; Aemisegger et al., 2015). The existence of drop
evaporation is further supported by a relation between Δd-excess (r-v) and surface relative humidity (RH;
$R^2$=0.31; Fig. 4c). The difference between rain and vapour isotopes is more in lower RH and less in higher RH,
as expected (Stewart, 1975). A similar analysis (Xing et al., 2020) in China also found that the change in
isotopic composition is large when RH is less than 60 %.

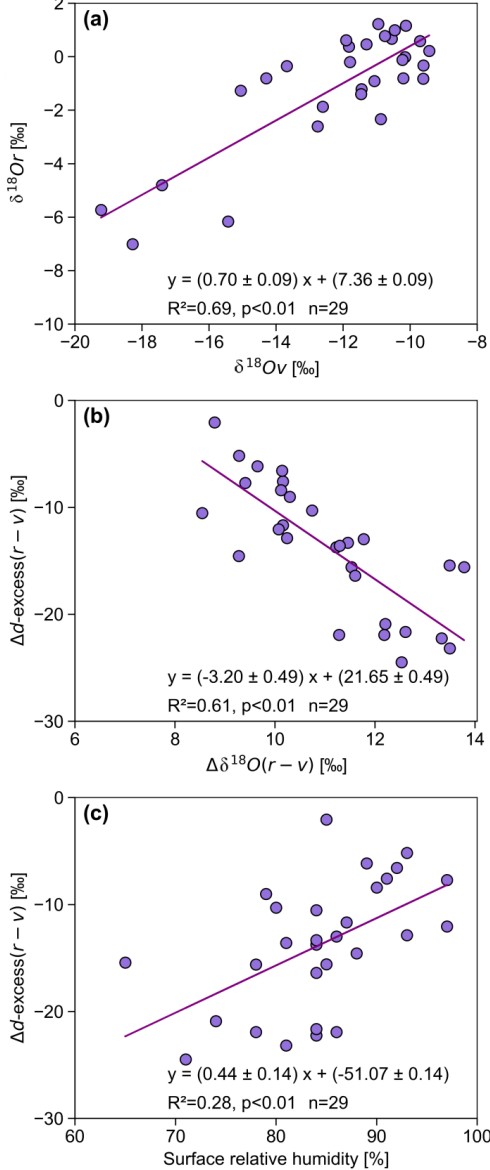






**Figure 4.** The correlations between **(a)** $\delta^{18}$O of rain ($\delta^{18}$Or) and $\delta^{18}$O of vapour ($\delta^{18}$Ov) at the ground level; **(b)** the difference in d-excess of rain and vapour ($\Delta$ d-excess(r-v)) and $\delta^{18}$O ($\Delta\delta^{18}$O(r-v)); **(c)** difference in the d-excess of rain and vapour ($\Delta$ d-excess (r-v) ) and ground level relative humidity (RH).

Falling raindrops and the water vapour in the atmospheric column constitute an interacting two-phase system, especially below the cloud base. On the way down, the water molecules are constantly exchanged between these two phases depending on the ambient RH and temperature. This makes the system evolve towards an isotopic steady state. The difference between isotopes of vapour in equilibrium with raindrops and the observed vapour (at the ground level, defined as $\Delta\delta$ and $\Delta$d) is useful to quantify the departure from equilibrium. Graf et al. (2019) demonstrated the importance of a $\Delta\delta$-$\Delta$d plot to represent the effect of sub-cloud processes, such as evaporation and equilibration, which influence the water isotopes. In our case, the expected equilibrium vapour isotope values were estimated by using the standard fractionation formula (Horita and Wesolowski, 1994) at the ambient temperature. The time series of $\Delta\delta$ values (Fig. 5a) for the Pune precipitation samples varied between -20 ‰ and 20 ‰ (omitting one outlier) respectively. For $\Delta$d, the time series shows negative values in all cases (ranging from 0 to -20 ‰). The close-to-equilibrium samples correspond mostly to the high-humidity period in July (Fig. 5b). Fifteen samples indicate the influence of below-cloud evaporation with positive $\Delta\delta$ values associated with strongly negative $\Delta$d values (up to -20 ‰).

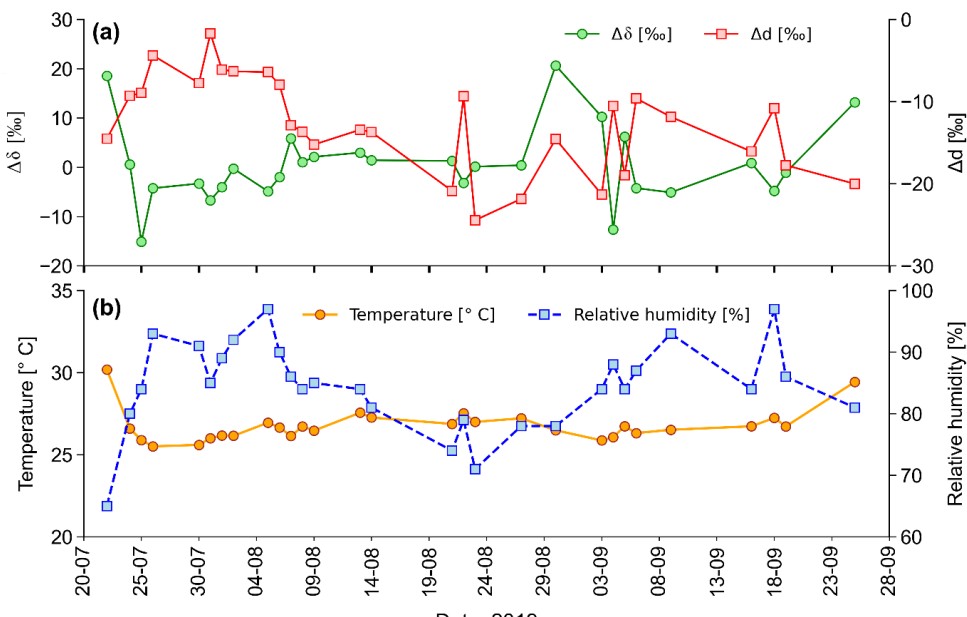

**Figure 5. (a)** Time series of $\Delta\delta$ and $\Delta$d of the rain samples collected during 2019 monsoon (July to September) in Pune. $\Delta\delta$ and $\Delta$d values (total points=29) as defined in the text following Graf et al. (2019). **(b)** Time series of daily average surface temperature and relative humidity recorded at IMD Pune observatory during the study period.



A Δδ-Δd scatter plot based on these observed data (Fig. 6d and 6h and Fig. 7d and 7h show that none of the rain
samples is in equilibrium with the corresponding ground-level vapour. About 63 % of the sample pairs fall in
the lower right quadrant of the diagram, where the raindrop evaporation is relatively more significant, as per
Graf et al. (2019). We note that the observed rainfall amount was low (less than 5 mm) for these samples, which
is consistent with a substantial evaporation effect. Nine samples have negative Δδ and Δd values, indicating
incomplete equilibration with near-surface vapour. The crucial driving factors for below-cloud processes seem
to be the size of raindrops and the intensity of precipitation. This is primarily because raindrops with larger
diameters correspond to increased intensity and have shorter residence times in the atmospheric column. As a
result, they experience reduced evaporation while descending toward the ground.
The regression line for Δd/Δδ has a slope of -0.43 (Fig. 6d). This is more than the slope of -0.3 reported
by Graf et al. (2019) for their study area, Zurich, Switzerland. Their study was based on short-time intra-event
samples in a mid-latitude region, whereas daily samples in a tropical region are used in the current study. A set
of complex processes operates to dictate the value of the slope, and Graf et al. (2019) pointed out that the slope
could represent a balance between below-cloud evaporation and equilibration of rainfall. They suggested that it
would be insightful to explore the slope for other climatic regions, hinting that the slope will help assess the
evaporation magnitude.
From the analysis of our data, it seems that drop evaporation is more important in our case (for the
same change of Δδ, the change in Δd is comparatively bigger). Simulation experiments by Graf et al. (2019)
showed that at high humidity (lower evaporation), the change of Δd is negligible, leading to a lower slope.
Conversely, when the temperature is higher, the slope is higher due to higher evaporation. Below, we explore
how accurately the BCIM can simulate rainwater isotopes in our tropical location.

**4.3 Application of BCIM with appropriate input parameters**
**4.3.1 Setting the boundary condition of the model**

As mentioned before, to estimate hydrometeor isotopic composition, the BCIM requires vertical profiles of
temperature, humidity and vapour isotope as input parameters. The vertical profiles can be introduced into the
model in two ways: (1) Vertical ascent assumption. Here, the profiles can be calculated based on an idealised
Rayleigh model having moist adiabatic ascent of air parcels from the surface to the top of the column, and (2)
the T, RH profiles can be constructed based on available sounding data and isotope profiles can be derived from
simulations conducted using isotope-enabled atmospheric models (Pfahl et al., 2012). These are discussed below
(Sections 4.3.2 and 4.3.3).

**4.3.1.1 Formation height and drop size assignment**

The formation height of the drop is an important factor and should be fixed by considering the most probable
altitude range. This parameter is not known a priori, but we can infer this from the cloud liquid water content
analysis. An earlier study by Kumar et al. (2014) pointed out that a peak of Cloud Liquid Water Content
(CLWC) is often present at 850 mb during the monsoon season over western India. The CLWC data for a period
of 29 days of the study period obtained from the ERA-5 dataset also show a peak at 850±50 mb (Supplementary



Fig. S3). Here, we consider the CLWC peak at 850±50 mb (about 1 km above ground from Pune) as the drop
introduction height for our case, where the RH reaches the value of 100 % (following Graf et al., 2019).

The BCIM also requires an initial drop size at the formation height. Unfortunately, no disdrometer or

MRR observations are available in the study area during 2019. We, therefore, adopted an empirical procedure,
known as the Marshall-Palmer relationship, to estimate the mean drop size at the ground. This was done by a
weighing procedure. First, we estimated the hourly mean drop size of the raindrops at the ground level from the
hourly rain rate data available from an IMD observatory at Shivajinagar, Pune, located about 4 km away from
our study area. Next, we calculated the 24 h mean drop size by taking a weighted average of the size and using
rain rates as the weights. The surface drop sizes thus calculated vary from 0.61 to 1.80 mm for various days. The
drop diameter at the ground is next provided as an input, and then the initial size at the drop height (about 1.5
km above ground) is estimated iteratively in BCIM using the microphysics part of the model. This procedure
was adopted for each day.

**4.3.2 Results of simulation**
**4.3.2.1 Run-1: Rayleigh ascent assumptions**

As mentioned above, the model needs vertical background profiles of atmospheric temperature (T), relative
humidity (RH), $\delta Dv$, and d-excess, dv. In Rayleigh simulations, various profiles were calculated from the moist-
adiabatic ascent of an air parcel with surface values of temperature (T0), relative humidity (h0), $\delta Dv$ ($\delta v,0$) and
dv (dv,0) of each sampling day as inputs (see isotope profiles in Supplementary Fig S4a and b). The surface
values of $\delta D$ and d-excess of vapour were taken from our vapour measurements along with the daily
temperature and humidity data obtained from the IMD publication (Section 2.3). The results for the set of
calculations using the Rayleigh ascent assumption (designated as Run-1) are shown in Fig. 6(a-d). In this set of
figures, we compare observed and model rain $\delta D$ (Fig. 6a), $\delta^{18}O$ (Fig. 6b), and d-excess (Fig.6c) values. We
also construct $\Delta\delta-\Delta d$ diagrams for both observed and model values and compare them in Fig. 6d. Although
observed and model isotope values (Fig. 6a and 6b) show strong correlation ($R^2$=0.86 and 0.79, respectively),
the model values are mostly overestimated (the plotted points lie below the 1:1 line). The overestimations of
isotopes (for $\delta^{18}O$ and $\delta D$) affect the d-excess values considerably more; the points lie far to the right, and no
correlation exists between the observed and model d-excess values (Fig. 6c). This is because the d-excess
parameter is more sensitive to departure from equilibration due to dominance of evaporation, which means that
a small departure of delta values would magnify the discrepancy in case of d-excess. We also note that most of
the model data points in a $\Delta\delta-\Delta d$ cross-plot do not agree with the observed ones. However, they do fall in the
lower right quadrant, which is consistent with high raindrop evaporation. We also note that the $\Delta\delta$ and $\Delta d$ model
values (Fig. 6d, Run-1) show smaller variations compared to the observations. The $\Delta\delta D$ of the model
simulations varies from 0 ‰ to 5 ‰ and $\Delta d$ from 0 ‰ to -5 ‰, while the observed values have variations of
about 25 ‰ (higher by a factor of 5). These comparisons show that the Rayleigh ascent model fails to reproduce
the evolution of the rain isotopes in our region.

**4.3.2.2 RH and T from Radiosonde and isotope profiles from TES and LMDZ (Run-2)**



Rayleigh ascent in Run-1 assumes that the source of vapour aloft is the rising air parcel, and the isotope values
along with RH and T should reflect that. But this did not yield a good fit. The simulation can possibly be
improved if we use RH and T data from local radiosonde observations and different isotope profiles. For the
present period, the radiosonde data were available only at a few specific pressure levels, and hence, appropriate
interpolations were carried out. To obtain the vertical profiles of vapour isotopes, we first use the isotope
outputs of a GCM, LMDZ for Pune (Dr. Camille Risi; personal communication). These values are used in
BCIM as inputs, and the simulated rainwater and vapour composition were compared with the observed values.
We found that a wide difference exists between the observed and model rain/vapour isotopic values. We suspect
that the LMDZ model may not be able to simulate the vapour isotope ratios accurately. This limitation was
noted by Risi et al. (2021) in a recent study involving large-eddy simulation; they observed that for high
precipitation areas, the convective or mesoscale downdrafts bring more depleted vapour from above into the
sub-cloud layer. Therefore, as an alternative, we used the δDv profiles modified from TES observations. These
profiles are constrained by using the measured ground-level vapour isotope ratios as a boundary condition while
maintaining the shape of the profile. The procedure is discussed in the light of our analysis period.
Firstly, TES vapour δD data are not available for 2019. Moreover, it is also known that the data have
large uncertainty within the boundary layer (Nimya et al., 2022). This necessitates the derivation of vapour
isotope profiles, which would merge with the TES observations at upper layers. The TES provides δD values of
moisture at 17 pressure levels with a 5.3 km × 8.4 km footprint during the years 2005-2009. Based on these, we
derived an average TES profile, which is deemed to be representative of the mean monsoon values constructed
by averaging TES observations over a box (16°-20° N; 72°-76° E) for the ISM period.



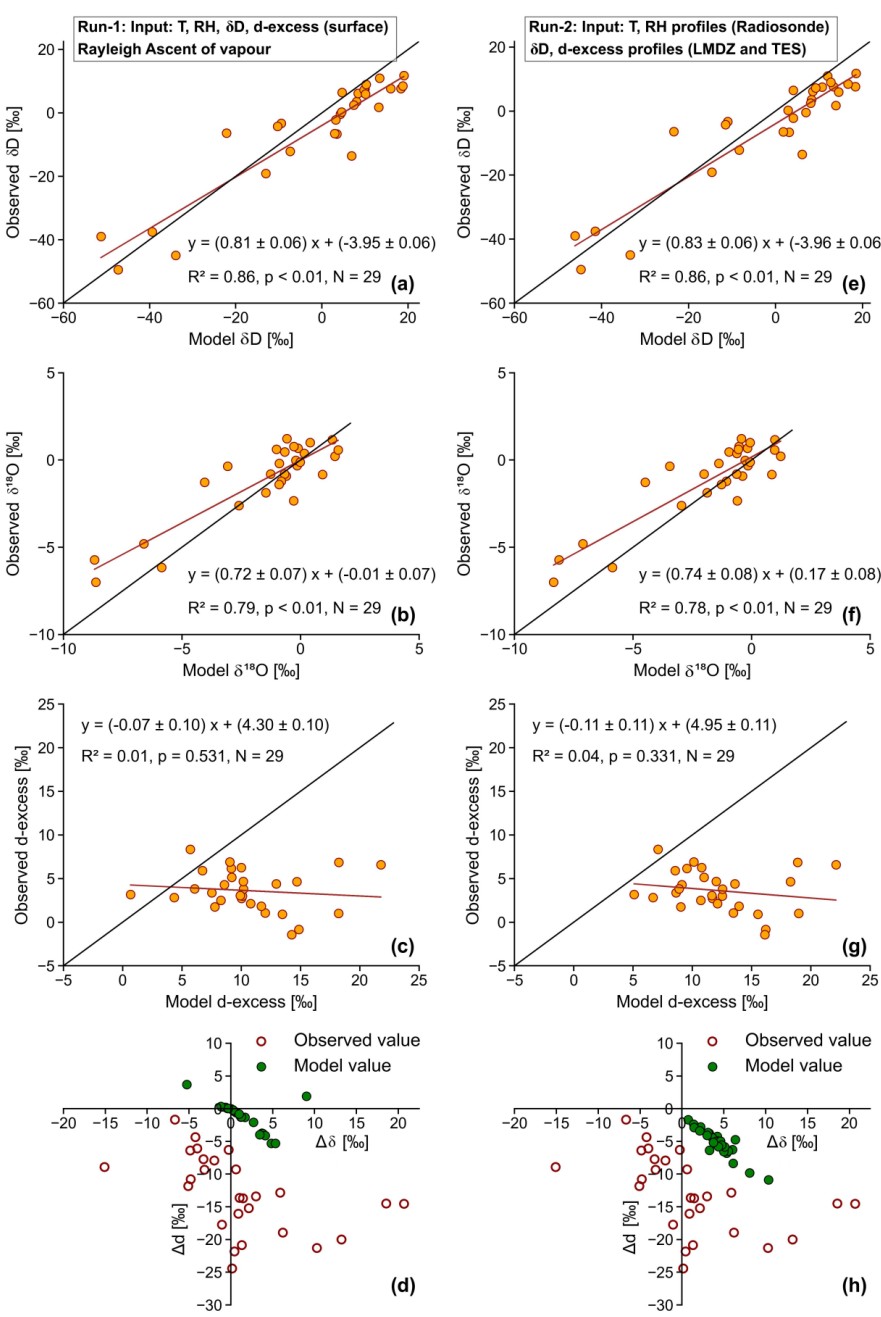

**Figure 6.** Scatter plot showing observed and simulated **(a)** rain δ$^{18}$O, **(b)** rain δD, **(c)** rain d-excess, and **(d)** same data in Δδ−Δd diagram. Rayleigh ascent of a surface air parcel is assumed here (named as Run-1). Similar plots for Run-2 are shown in **Fig. 6e-h**, in which input profiles of T, RH, and vapour δD and d-excess values are obtained by adopted TES and LMDZ outputs (see text).



To derive the vertical profiles for each of our sampling days, we use our daily surface measurements as boundary values. The average TES profile (as mentioned above) is modified through a curve-fitting technique where the shape of the average profile is slightly altered while being constrained to pass through the surface value. A 4th order Polynomial of the type: $Ah^4 + Bh^3 + Ch^2 + Dh + E$ (where h is the altitude in meters) was fitted to the average profile after adjusting its surface value so that a smooth shape is obtained (D/H decreasing with height following the average pattern). The polynomial coefficients (five in number) were calculated for three cases: (1) for the maximum observed surface D/H value, (2) for the mean surface value, and (3) for the minimum observed value, giving us three sets of A, B, C, D and E values. The constants for each day were next estimated by interpolation using these three sets. Obviously, this method of interpolation, constrained by surface vapour measurements, assumes that the vapour aloft is related to the surface value, and this assumption may not be correct. But it, at least, allows us to check if the surface constraints yield better rain isotope ratios at the ground (using BCIM) while being consistent with the TES measurements of vapour aloft.

Unfortunately, the vapour $\delta^{18}O$ values at various pressure levels are not available from TES (which gives only the HDO/H$_2$O ratio). Therefore, we adopted a derivation technique using the vapour isotope profiles simulated by the LMDZ model. In this technique, daily average vapour $\delta^{18}O$ and $\delta D$ values were obtained from the LMDZ model outputs over our study region for the sampling dates at each height. For each day, two profiles (for $\delta^{18}O$ and $\delta D$) were constructed, and polynomials were fitted. Next, the d-excess profiles were constructed from these two profiles. Each of the daily d-excess profiles was then constrained by using the surface d-excess vapour value for that day to obtain the fitted d-excess Polynomial for that day. The rationale is that even though the individual profiles of $\delta D$ and $\delta^{18}O$ provided by LMDZ do not predict well the rain isotope ratios (as seen by our trial), the d-excess based on these two isotope ratios should be reasonably good. The obtained vapour d-excess and $\delta D$ profiles are shown in Fig. S4c and S4d, Run-2. These profiles were subsequently employed in BCIM (named Run-2) to generate the daily-scale $\delta^{18}O$, $\delta D$ and d-excess values of surface rain isotope ratios (Fig. 6e-6h). However, the results do not show much improvement compared to the Run-1 (Fig. 6e-g) despite showing a larger variability in the $\Delta\delta$–$\Delta d$ plot (Fig. 6h); the $\Delta\delta$ values varied from -4.7 ‰ to 11 ‰ and $\Delta d$ from -1.8 ‰ to -12.4 ‰. Additionally, in this case, all the data points fell in the 3rd quadrant of the $\Delta\delta$–$\Delta d$ cross plot (Fig. 6h and Fig. 6d). Both Run-1 and Run-2 simulations fail to yield a good match between the observations and model (especially the d-excess) values.

### 4.3.4 Vapour $\delta^{18}O$ correction in the profile (Run-3 and Run-4)

The main source of error in Run 1 and Run 2 could be improper vapour isotope profiles. It is possible that the true profile for a given date may not coincide with the surface-measured value in extrapolation, as assumed by the boundary constraint. In other words, the vapour aloft may not be derived entirely from the surface vapour as measured at our sampling location. One possible explanation could be a significant contribution from the small-scale local surface moisture having a different isotopic composition (evaporation or evapotranspiration from water bodies or trees within a few hundred meters). However, this possibility can be ruled out as a study using satellite data showed that due to high humidity and low temperature during ISM, evaporation/ evapotranspiration (~0.5 mm day$^{-1}$) adds a negligible amount of moisture compared to the advective fluxes in this region (Pathak et al., 2014). Our investigation is also limited by the absence of upper air





δD, $\delta^{18}$O values from an independent observation or model on a daily scale. Due to this limitation, we adopted a
forward modelling approach.

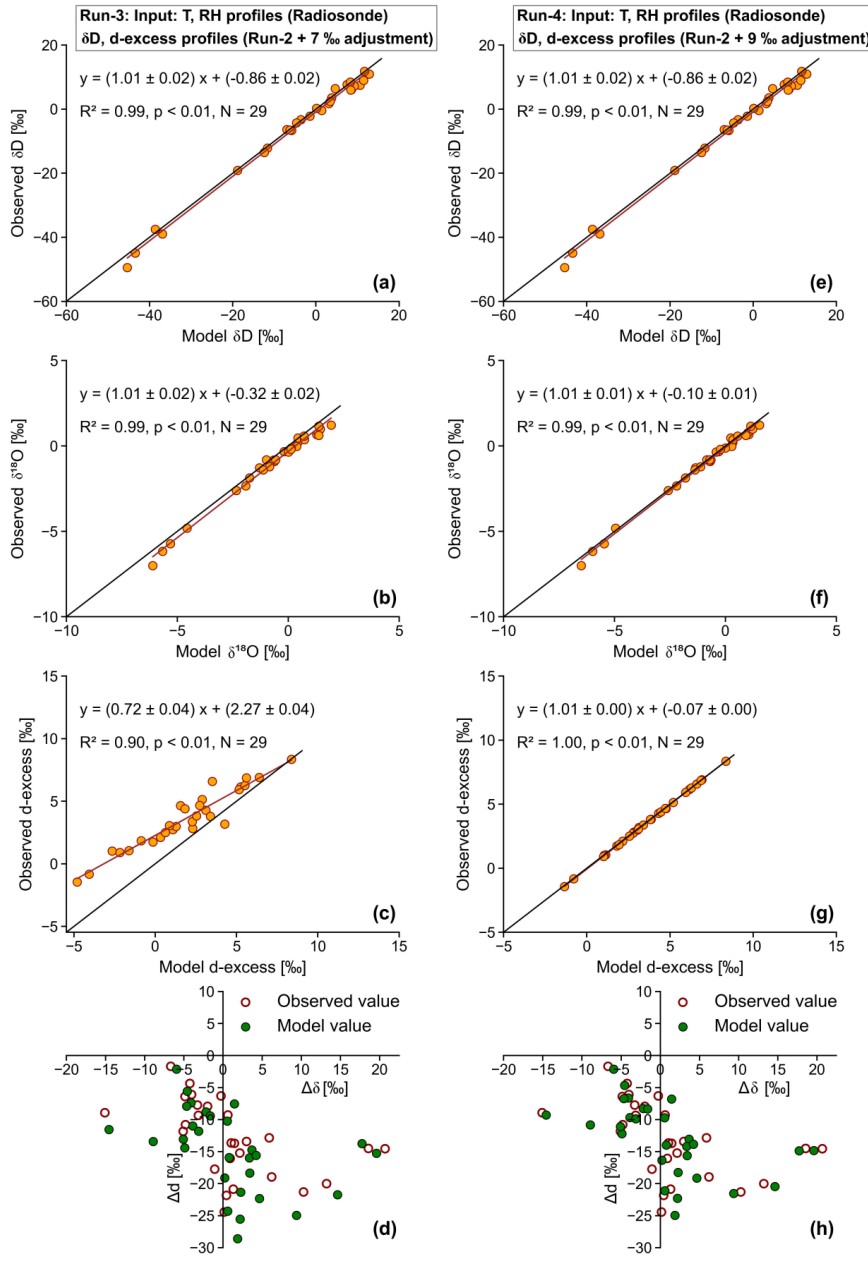


**Figure 7.** Scatter plot showing observed and BCIM simulated **(a, e)** rain δD, **(b, f)** rain $\delta^{18}$O, **(c, g)** rain d-excess; in **(d, h)**
data are cast in the form of Δδ–Δd diagram (for definition of Δ values see text). In the panels (a, b, c and d), the input vapour
$\delta^{18}$O values in the profile are reduced at each level appropriately so that maximum reduction is ~7 ‰ at the fall height; in the





panels (e, f, g and h) the reduction is ~9 ‰.

Keeping the D/H ratios nearly the same, we tuned the vapour $\delta^{18}$O input profile to achieve a reasonable
agreement for each date. Two such tunings are attempted. In both, we reduced the $\delta$D values slightly and
increased the $\delta^{18}$O moderately while keeping the shapes similar to Run-2. In Run-3 (Fig. 7a-d), the vapour $\delta^{18}$O
value is increased at each interval in such a way that the d-excess of the drop decreased to ~8.2 ‰ (on average)
from the measured surface value of about 17 ‰ (on average). In the second trial, Run-4 (Fig. 7e-h), the d-excess
decrease was made slightly less (average d-excess ~10.7 ‰). These changes are shown in the vapour isotope
profiles given in supplementary Fig. S4(f) and S4(h).
We recognise that it is difficult to validate the vapour $\delta$D or d-excess profiles constructed by the above
method due to a lack of height-specific observations. However, the available aircraft-based vapour isotope
observations suggest that both d-excess and $\delta$D values of vapour decrease with altitude and thus provide some
evidential support to the assumed decrease (Sodemann et al., 2017). With the above choices, simulations of rain
isotopes improve (Fig. 7) considerably (both in terms of the uncertainty of the slope of the regression line and
the correlation coefficient). Between the two alternatives of Run-3 and Run-4, Run-4 is found to be superior in
the matter of comparison of the model with observations; the average $\Delta$d (observation-model) difference
decreases from 2.1 to 0.4. Additionally, there is considerable improvement in the $\Delta\delta$-$\Delta$d cross plot (see Fig. 7d
and 7h).
The tuning exercise suggests that the adoption of the $\delta^{18}$O profiles or the d-excess profiles based on
TES $\delta$D and LMDZ $\delta$D/$\delta^{18}$O values (Run-2) was slightly in error. We found that, on average, the adopted $\delta^{18}$O
should be increased by about 0.4 ‰, and the adopted $\delta$D decreased by about 3.5 ‰. Consequently, the model d-
excess should be changed on average by about -7 ‰ (ranging from +3 ‰ to -17 ‰). A preliminary inspection
has shown that the situation would not improve had we taken another isotope-enabled GCM, IsoGSM2
simulation (instead of LMDZ) for $\delta^{18}$O-calibration; in fact, IsoGSM2 simulates higher values for both the
vapour isotopes ($\delta$D and d-excess; results not shown).

**4.3.5 Sensitivity analysis and uncertainty estimates**

In the context of point #4, we examine the effects of the formation height on isotopes, keeping all other
parameters the same. We increase the formation height by 1 km (from 1.5 km to 2.5 km) and run the BCIM. To
form the drop at a higher altitude, we need to change the RH profile so that the RH=100 % level is reached at
the new height. A simplified RH profile is used by approximating the real profile with a straight line, where the
surface RH value is taken as one end member, and the 100 % level is taken at the new height. We found that the
simulated values of the rain isotope ratios did not change significantly, and similarly, the raindrop evaporation
fraction also did not change. We provide detailed uncertainty estimates of the model rain isotope values in
Supplementary Information (SI-1). The uncertainty values for $\delta$D$_{rain}$ = 3.5 ‰ and d-excess$_{rain}$ = 2 ‰.
We also did a detailed sensitivity analysis (see Supplementary information, SI-2) to study the effects of
variation in temperature, relative humidity, vapour isotopes, and drop size using the BCIM. These analyses
show that vapour isotope values, RH, Temperature and drop sizes are the dominant factors controlling the model
rain isotope ratios.



### 4.3.6 Estimate of raindrop evaporation

Our analysis shows that with minor tuning of vapour profiles, the BCIM can be used to simulate the rain isotope ratios in Pune. For tuning, the d excess of the vapour needs to be reduced, on average, by about 7 ‰ compared to the observed ground vapours. Assuming the validity of this tuning, we find that the rains suffer substantial but variable evaporation in the Pune region. We see from the output of BCIM that the mass of the drop reduces as it falls. The ratio of final mass/initial mass (or remaining fraction of mass of the hydrometeor relative to the initial mass, i.e., $m/m_o$) can then be used to estimate the mass loss suffered by the drop on its way down for each day. The difference $(1-m/m_o)$ of the drop then represents the effective rain evaporation. Defining rain evaporation in this way, a time series of evaporation values is displayed in Fig. 8a, which varies from 4 % to 73 % (average ~23 %, omitting one outlier). As expected, drop evaporation is inversely related to the surface humidity (Fig. 8b) and drop diameter (Fig. 8d) but directly proportional to the temperature (Fig. 8c).

The evaporation was relatively high (59 % and 73 %) for two days (22 July and 21 August) when humidity was low (65 % and 74 %), and the temperature was high (30° C and 27° C) along with drop diameter being small; the combined effect resulted in high evaporation fraction (Fig. 8a). In general, the deduced evaporation fractions are high (23±16 %) in this region. This inference is consistent with the observed anti-correlation between d-excess and $\delta^{18}O$ of rain samples (Fig. 3b) as expected in drop evaporation when d-excess of the raindrop decreases while its $\delta^{18}O$ increases.



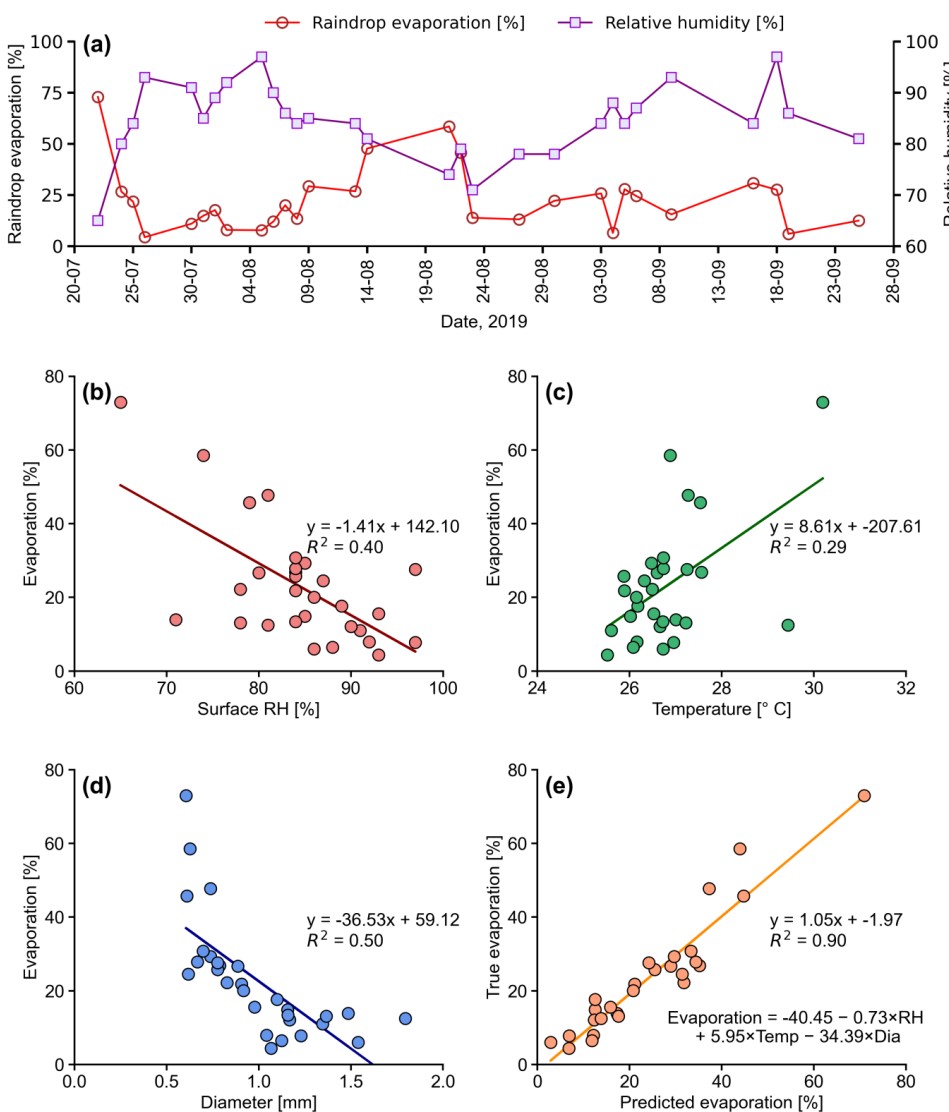

**Figure 8. (a)** Time series of raindrop evaporation estimated from the BCIM using the simulation in Run-4 and surface relative humidity. The regression between raindrop evaporation with **(b)** RH, **(c)** temperature, and **(d)** drop diameter. **(e)** Multiple regression analysis yields a joint equation: Evaporation (%)=-40.45-0.73*RH+5.95*temperature-34.39*drop diameter

A multi-variate regression analysis shows that we can fit the evaporation fraction (in %) as a function of three surface variables: RH (%), temperature ($^{\circ}$C) and drop diameter (mm) as below (Fig. 8e):

Evaporation Fraction= 40.45-0.73*RH+5.95*Temperature-34.39*Diameter    ($R^2$=0.88)    (1)

As we see, these three parameters control the total variance of the error and among them, the RH is the




major one because the observed temperature does not vary much (26.8±1.0 °C), being only about 4 %, while for
RH, the variation is larger (84.5±7.2 %) on the order of 8.5 %. The diameter variation is also rather small
(1.0±0.3 mm). From the above relation, we estimate an uncertainty of ±10 % for the model evaporation fraction
(assuming errors of 5 % in RH, 0.5° C in T, and 0.3 mm in diameter).

To explore the influence of drop evaporation on rainfall amount, we plot evaporation as a function of

rainfall in Fig. 9 which shows that the two parameters are related by a power law where an increase in the drop
evaporation causes a reduction in the rainfall. However, for large rainfall, the evaporation influence is less; for
smaller rainfall range (less than 5 to 10 mm/day), the evaporation change affects the rainfall significantly. For
example, even for a minor increase in evaporation, say from 20% to 30%, the rainfall decreases from 2.1 to 0.5
(mm/day). The reason is that smaller rainfall is usually associated with smaller drops which suffer relatively
more evaporation, considering other parameters (RH, Temperature) constant.

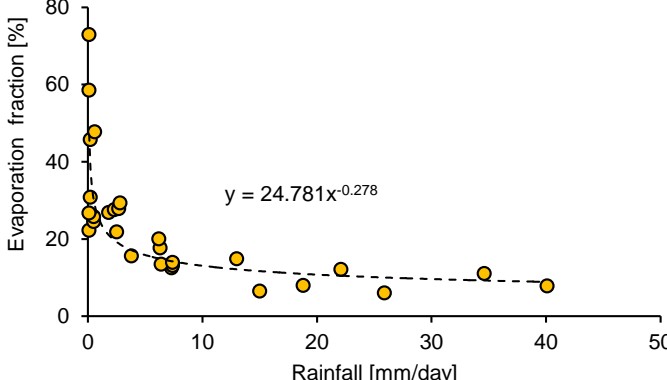


**Figure 9.** Scatter plot showing relationship between the drop evaporation estimated in this study and rainfall in Pune. The
black dashed line indicates the best-fit power law.

**5. Summary and Conclusions**

We analysed isotope ratios of daily rain and atmospheric vapour samples collected from surface level

at Pune, a tropical rain shadow region in Western India, during the summer monsoon season (June-early
October) of 2019. The key findings are listed below:

1. The vapour isotopes show considerable temporal variation (with $\delta^{18}O$ from -19.2 ‰ to -9.4 ‰ and

$\delta D$ from -123.7 ‰ to -63.4 ‰). Among the diversity of variations, there were four events

extending over a few days when both rain and vapour isotope ratios were considerably lower (for

example, rain values were less than the mean -0.5 standard deviation; with $\delta^{18}O < -2.6$ ‰). These

events seem to indicate intimate relations with regional meteorological characteristics.

2. We note that the low rain isotope events are found to be synchronous with negative OLR

anomalies. Negative OLR anomalies in the tropical monsoon zones of India are known to be

associated with large-scale convections which uplift air masses to great heights (Sengupta et al.,

2020); We surmise that in such cases, the rain formation takes place in an environment of cold



temperatures (larger fractionations) and low δ-values of ambient vapour. Both these factors would
yield low rain δ-values.
3.   A gradual increase in the d-excess values of vapour and a small but notable decrease in $\delta^{18}$O
values in the later part of the monsoon (after mid-August) were noted. The very high vapour d-
excess in September is especially noticeable. In contrast, the rain d-values are not significantly
different. We also find a strong anti-correlation between vapour $\delta^{18}$O-d-excess values.
4.   The above observations suggest increased moisture recycling in the form of vapour contribution
from evaporation of raindrops and/or local vapour supply. However, local-scale vapour supply
cannot be a large factor based on an earlier study in central India (Pathak et al., 2014). Therefore,
we strongly believe that downdraft of depleted vapour is the main source of low isotope (and high
d-excess) surface vapour (Risi et al., 2023). The depleted vapour in the sub-cloud region can
originate from raindrop evaporation.
5.   To quantify the sub-cloud processes altering the rain isotope values, we used the Below Cloud
Interaction Model BCIM. Upon reasonable tuning of the input parameters, we obtained a notable
agreement between the observed and model rain isotope values at the ground level.
6.   In the Δδ-Δd (Δ is defined by rain-equilibrium vapour minus the ambient vapour following Graf et
al) cross plot, the majority of the data points lie in the 3$^{rd}$ quadrant, which signifies the dominance
of raindrop evaporation over Pune and the adjoining region during our study period. The cross-plot
is indicative of drop evaporation but cannot quantify the magnitude. The slope of the points (about
-0.43), however, suggests that evaporation is intense. This is because a higher slope in the cross-
plot is caused by a relatively magnified effect of d-excess difference between the rain (and
corresponding equilibrated vapour) and the ambient vapour which is due to a larger evaporation.
For reference, Graf et al. (2019) found that the slope was lower at a value of -0.3 for Zurich.
7.   Since the BCIM is found to be applicable to our study area, we estimate the raindrop evaporation
parameter from the model output. An event-to-event quantification of raindrop evaporation is the
key finding of our study. The model gives a net reduction of the drop mass at the ground level, and
we can define the relative reduction as a measure of the effective rain evaporation. Using this
innovative technique, the model shows that, on average, about 23 % (varying from 4 % to 73 %) of
the rain evaporates in the sub-cloud layer. There are four abnormally large values (46, 48, 58, and
73 %) of evaporation. The largest value is probably due to low RH (~65 %) on that day, but as for
the other days, probably a combination of smaller drop size and lower RH played a role. Excluding
these four values, the average evaporation is 18±8 % (range of 4 to 30 %).
It is instructive to compare our results to the evaporation estimates obtained in similar studies carried
out in other climatic regimes. Sarkar et al. (2003), in a steady state one-dimensional model study of rain in the
North Atlantic Trade Wind region (Barbados), found a high value of 63% (63±23 %) for raindrop evaporation
which is three times more than our average value of 23% (23±16 %). The reason for this is a large difference in
drop size and RH. A comparison reveals that their drop size was much smaller (from 125 mm to 6 mm) in
comparison to ours (from 0.61 to 1.80 mm). The drops were so small (smaller than 300 mm) in some cases (4
February,2020), that they completely evaporated (evaporation ~ 100%) during the fall leading to very small
rain. In addition, in their sampling region, the RH was also lower, ranging from 65% to 80%, compared to ours



(65% to 97%). Lower drop size and lower RH lead to higher raindrop evaporation. In addition, the drop sizes
varied over a larger range, leading to a larger variability compared to our study.
In another study, rain and vapour isotopes were measured in a cold-front passage over Zurich during
19-25 July 2011, and the data were interpreted by an isotope-enabled regional weather prediction model
COSMOiso (Aemisegger et al., 2015). The authors showed that by switching off the raindrop evaporation, the
rainfall increased by about 75% because the cooling induced by evaporation causes diminished convective
activity. The estimated average evaporation in their study was about 40% (Dr. F. Aemisegger, personal comm.).
This value is also twice our value. The reason is probably lower drop size and lower RH; as stated in their paper:
"weak rainfall intensities (small droplets and thus lower falling velocities), and the possibly lower relative
humidity in the air column above could have contributed to the evaporative enrichment of precipitation".
The tracer-based technique and the BCIM, which we used, are associated with a series of limitations-
a)  We used TES satellite data averaged over 2005-2009 to guide our choice of vapour isotope
profiles, but the year of analysis was 2019. In this matter, there is no way to ascertain the degree of
deviation of the true profile from the adopted ones in Run-2.
b)  The $\delta^{18}O$ profiles were adopted based on the $\delta D$ and $\delta^{18}O$ profiles obtained from the LMDZ
model. As noted, this did not give us good agreement with the observations.
c)  The isotope profiles were constructed using ground observations as boundary values. However,
this also resulted in a mismatch with the observed values, and we had to tune to lower $\delta^{18}O$ values
and higher d-excess values to achieve good agreement. It should be mentioned here that Risi et al.
(2023) also discussed a similar idea in their study of water isotopes in tropical squall lines, that
convective downdrafts can introduce depleted vapour produced by rain re-evaporation in the
boundary layer. Moreover, the vapour samples were collected for a duration (about a few hours)
that did not coincide exactly with the longer rain collection period (about 24 hours).
d)  The raindrop formation height was assumed to be the same for all rainy days, and the drops were
all introduced at a constant level, considered to be the cloud base at RH=100 %. However, it is
well known that raindrops do not all form at the same height, even on a single day. With this
assumption, we are neglecting alterations in isotope ratios produced inside the cloud by various
microphysical processes. However, since we are concerned with sub-cloud processes, this is not a
serious problem.
Considering these limitations, we provide detailed uncertainty estimates of the model rain isotope values in
Supplementary Information (SI-1) and raindrop evaporation estimates (Section 4.3.6). The uncertainty values
for $\delta D_{rain}$ = 3.5 ‰, for d-excess$_{rain}$ = 2 ‰, and drop evaporation estimate is 10%.
Presence of evaporation during ISM has been postulated earlier in several theoretical models, but this
study provides, for the first time, a quantitative estimate of rain evaporation on a day-to-day basis in the Indian
monsoon season using combined rain vapour isotope data. However, a ~25 % raindrop evaporation applies only
to the highly humid Pune region. The average seasonal rainfall in Pune is about 55 cm (for ISM), and if ~25 %
of this is evaporated, it would mean considerable cooling of the boundary layer leading to localized downdrafts,
formation of cold pools, and changes in atmospheric stability. The cooling can also hinder efficient formation of
convection (Hwong and Muller, 2024) and can have a large effect on the precipitation patterns in the tropics
(Bacmeister et al., 2006; Sarkar et al., 2023). Given the large share of precipitation recycling found in this study for



Pune, the question arises, how large precipitation recycling is at larger scales, i.e., regional or continental scales, as
well as in other seasons over India. We need to have a comprehensive program for carrying out such analysis,
aided with appropriate BCIM input parameters, to understand the evaporation of raindrops over various climatic
subdivisions in India. Moreover, high-frequency observation of vapour and rain isotopes would be useful to
quantify this fraction during various convective events associated with low-pressure systems during ISM. As
mentioned above, raindrop evaporation is an important parameter in modelling the energy and moisture budget
in monsoon rainfall prediction.

**Data Availability**

Observed rain and vapour isotope data are available upon communication with the corresponding author. The
upper-air radiosonde measurements were obtained from the University of Wyoming repository
(http://weather.uwyo.edu/upperair/sounding.html). The daily gridded data (zonal and meridional wind, specific
humidity, air temperature, and cloud liquid water content) are available from the European Centre for Medium-
Range Weather Forecasts Reanalysis (ERA-5; https://www.ecmwf.int/en/forecasts/datasets/reanalysis-
datasets/era5). The rainfall data (cumulated over 24 hours) are obtained from the Pune observatories of the IMD
(available at the National Data Centre (www.imdpune.gov.in/ndc_new/ndc_index.html)). Apart from daily
rainfall, hourly rainfall data and daily average temperature and relative humidity data for the Pune observatory
were also obtained from the IMD using the above link. The datasets for 48 h air mass back trajectory analysis at
850 mb pressure level are obtained from the NOAA Hybrid Single-Particle Lagrangian Integrated Trajectory
(HYSPLIT) model (https://www.ready.noaa.gov/HYSPLIT.php). We received daily outputs of LMDZ isotope-
enabled GCMs, which were provided by Dr. Camille Risi by personal communication. The Interpolated
Outgoing Longwave Radiation (OLR) data from NOAA
(https://psl.noaa.gov/data/gridded/data.olrcdr.interp.html) is used in this study. Tropospheric Emission
Spectrometer (TES) Level 2 (Nadir-Lite-Version 6) retrievals of HDO and $H_2O$ profiles for the available period
(2005–2007; https://tes.jpl.nasa.gov/tes/data) are used to construct the vapour $\delta D$ profile.
**Author Contribution**

SSN carried out all rain and vapour isotopic measurements and part of the data analyses, installed and ran the
model BCIM. SPR analysed the majority of the isotopic data, performed all controlled runs in the BCIM, and
constructed most of the figures. SS conceptualized the scientific plan and methodology and wrote the initial
draft of the manuscript. SKB contributed to data analysis and interpretation of model outputs, corrected the
manuscript, and provided useful comments and suggestions.

**Code Availability**
We carried out data analysis and plots using licensed versions of Microsoft Excel and Python, the latter being
freely available from https://www.python.org/downloads/. The code of the model, BCIM, is freely available
from https://git.app.uib.no/Harald.Sodemann/bcim.

**Competing interests**
The authors declare that they have no conflict of interest.



**Acknowledgements**
The Indian Institute of Tropical Meteorology, Pune (IITM), is fully supported by the Earth System Science
Organization (ESSO) of the Ministry of Earth Sciences, India. This work forms part of the Ph.D. thesis of SSN,
who thanks IITM for a fellowship. SPR thanks IITM for a research associateship. We thank Director IITM for
his constant encouragement. The NASA Langley Research Centre and the Atmospheric Science Data Centre are
acknowledged for the TES dataset. A fruitful discussion with Dr. Camille Risi is also acknowledged.

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
