# Peer review of "Assessing raindrop evolution over northern Western Ghat 2 from stable isotope signature of rain and vapour"

_EGUsphere, 2025_

## Referee Comment (RC2)

**"Assessing raindrop evolution over northern Western Ghat from stable isotope signature of rain and vapour"**

**Summary:**

The investigators aim to assess the degree of evaporation of rain between cloud base and ground level by linking ground based water isotope measurements with a below cloud isotopic model.

Ground based samples of rainwater and ambient vapor, located at the Indian Institute of Tropical Meteorology in Pune, are collected throughout the 2019 summer monsoon season. It appears that rainwater was collected on a 24 hr basis. Vapor was collected by pumping air through a chilled vessel to condense moisture out of the air, with a time resolution of 3-4 hours. Vapor and rainwater were sometimes collected on the same day, and are collocated.

The water isotopic trend throughout the season is analyzed in the context of large scale meteorology. Using outgoing longwave radiation anomaly (from satellite), a link between mesoscale convection and depletion of rainwater isotopes is demonstrated. Analysis of observed rainwater and vapor isotopes in  $\Delta\delta$ - $\Delta$ d space indicates the prevalence of below-cloud evaporation taking place to various extents throughout the season. Additionally, there are signatures in the vapor isotope data which lead the authors to believe that depleted vapor is being downdrafted potentially as a result of raindrop evaporation.

The authors then attempt to use the BCIM model to compare modeled raindrop isotopic evolution with the ground based measurements. Several estimates must be provided as inputs into the model: droplet formation height, droplet size, atmospheric temperature, humidity, and vapor isotopic profiles. The authors describe three approaches:

- 1. Using ground based temperature, relative humidity, and vapor isotopes, assume a moist adiabatic atmospheric lapse rate. The droplet begins at a fixed height of 850 mb. Droplet sizes are calculated from daily rainfall rates.
- 2. Radiosondes are used for thermodynamic profiles. Vapor dD is obtained from 2005-2009 satellite retrieval, while dO18 is obtained using d-excess profiles from a GCM and extrapolating from the observed vapor d-excess. Droplet sizes from daily rainfall rates. 850 mb droplet formation height.
- 3. Vapor isotopic profiles from method 2 are tuned to produce a d-excess at ground-level value in agreement with the observations.

The authors use the final tuned model input to conclude that on average 23% of rainfall during the season evaporated during descent.

**Overall assessment:**

This study presents a new set of collocated rain and vapor isotope observations, a measurement approach which holds great potential for addressing the topic of interest here. The simultaneous observations of vapour and precipitation (as opposed to only one or the other) is particularly advantageous. However, with the exploration of the BCIM that is

presented here, there is too much going on in this study, making the results muddy. This reviewer believes that the impact of and insights from the in-situ measurements is detracted by the troubled attempt to reproduce the observations with the BCIM.

This reviewer does not find the methods used to initialize the BCIM model to be convincing, and therefore holds little confidence in the quantitative values for evaporated fraction obtained from the model. While acknowledging and commending the work that the authors have devoted to the model analysis, this reviewer feels that in its current state it can be omitted entirely. It is therefore not ideal that the focus of the abstract is based on the BCIM results. A more impactful study might focus thoroughly on the observational dataset, strengthening the analysis in that area, and removing or heavily revising the modeling component.

Observational measurements are less common and highly valuable, and thus the sampling of rain and vapor isotopes is compelling. The authors have produced an observational dataset of high value, and the measurements are likely to provide a valuable and novel contribution to this research field. The authors can strengthen their findings from the observational data by further substantiating points #1-4 and #6 of the conclusions. More care should be put into describing the data collection methods and ascertaining firm conclusions on the observed trends within the context of other literature.

**Research Methodology:**

Below are specific comments on the research methodology, first for the experimental component, and then for the modeling.

**Observational measurements:**

The measurements in this study that are compelling:

- Rainwater isotope fraction and d-excess: I'm convinced that these measurements
  are highly reliable, although the authors should clarify if the rainwater is collected
  every 24 hours, and how this is done (e.g. what time of day is the sample
  collected?). Also, the link to the IAEA document is not easily accessible (I was not
  able to access it on a number of different web browsers and devices). So, it would be
  good to briefly summarize the collection method in case the reader can't access the
  link.
- Ground based rainfall rate, temperature, humidity from IMD: Although not precisely collocated with the rainwater isotope samples, the daily averages are temporally consistent with the rainwater collection and are likely valid.
- Radiosondes: Although not precisely collocated with the rainwater samples, and the mismatch in temporal resolution is severe (24 hr integration for isotopes, vs averaging of two point measurements for sondes), atmospheric profiles in the same region and date-range as the rainwater isotopes provide extremely valuable atmospheric context. I think the authors could be more careful with dealing with the temporal discrepancy. For example, on days with precipitation, it would be wise to verify that radiosondes were launched during the time of precipitation. As an extreme

example, if the day's precipitation accumulated between 04-09Z, but the sondes are launched at 00Z and 12Z, the atmospheric soundings for that day might be very misrepresentative, and may not even contain clouds. In general, I think these radiosondes could be utilized to a greater extent in the observational analysis.

**Vapor water isotope collection:**

- It's possible that these vapor isotope measurements are highly reliable, but more details need to be provided by the authors to allow the reader to make this assessment. It seems that a detailed description of the vapor collection apparatus has not been published prior, and no literature references were provided for the design. As such, including a picture or a diagram of the apparatus would be helpful. More importantly, were there any calibrations or validation of the apparatus performed prior to collecting the measurements?
- A question I have is about the level of certainty that the flow rate, temperature, and surface area within the chilled flask was sufficient to condense out all of the water vapor from the air sample as it's pumped through? If the water vapor is only partially removed from the sample (with the remainder flowing out), then significant kinetic fractionation could occur during the process of condensing the water vapor inside the dewar. If this were to be the case, the water condensed in the dewar would adopt an enriched bias, and an elevated d-excess that could also have a dependence on humidity. This may possibly explain some of the d-excess trends in the water vapor that were unaccounted for. Can the authors comment on this?
- More details on the timing of the vapor sample collection is also needed Were the samples collected once per day? Or during intensive observation periods?

The authors explain that the local advective vapor fluxes in this region are dominant, and any vapor from evaporating rain is likely negligible in the vapor isotopic signature. Could the authors assume a well-mixed condition below cloud base (assume that vapor isotope ratio is constant below cloud base)? In fact, this is the assumption that is made in the Rayleigh model: See Fig 4C of Graf et al 2019, where ambient vapor isotopes (black line) are constant up until the cloud base. A linear temperature profile from the radiosondes would also be indicative that a well-mixed boundary layer assumption is valid. The authors mention that the rainwater is not in equilibrium with the vapor at ground level. If the vapor isotope is constant below cloud base, and a temperature profile is known from radiosondes, the authors could identify (based on temperature) the height where the rainwater would be in equilibrium with the vapor. It would be informative to compare this to the lifting condensation level (aka cloud base height), and potentially some qualitative insights on isotopic exchange could be derived.

The interpretation of the slope in Line 402-403 and point #6 of the Conclusions section could be strengthened. Fig 7 of Graf et al 2019 shows different conditions that would lead to a steeper slope in the  $\Delta\delta$ - $\Delta$ d space. The authors have discussed the effects of RH and temperature on the expected slope: What about the ambient vapor isotope values? For example, if the ambient vapor d-excess in Pune is larger than the ambient vapor d-excess in Switzerland, then one would indeed expect the Pune rainfall data to fall along a steeper slope than that in Switzerland. From the observed data presented, it appears that Pune vapor d-excess values range from 10-30 while the data in Graf et al 2019 is in the 5-20 range, so this could be a potential source of the steeper slope. Likewise, higher vapor  $\delta$ D

values would produce a steeper slope, which again the Pune values appear to be larger than those observed in Switzerland. For completeness, all the different factors that might contribute to a steeper slope should be considered and discussed to strengthen this analysis.

Line 357: Is the ambient temperature used for the  $\Delta\delta$ - $\Delta$ d plots obtained from the IMD observatory?

Line 389: Can the authors comment on if the <5mm rain rate of the data in the lower right quadrant is relatively lower than those in the lower left quadrant?

Line 390: The text says nine samples, but the scatter plot shows 11 points in the lower left quadrant.

Line 391-392: The statement on size of raindrops and precip intensity - Is this a finding from your own observations? If so, please demonstrate how this was assessed.

**BCIM model:**

In general the authors should clarify if the radiosondes are being used for the thermodynamic profile in all of these cases (other than the Rayleigh case), or if the GCM, or satellite profiles are being used. Perhaps the authors could provide a table for each run, specifying the BCIM inputs and where they came from: droplet size, introduction height, Temperature and humidity profile, Deuterium, and O18. Or just be very clear in the description.

Showing plots of the atmospheric temperature and RH profiles that are being input into the model would be an enormous help to the reader for understanding/assessing the atmospheric structure being modeled. In addition to the isotope profiles shown in Figure S4.

If I understood correctly, run-2 uses radiosonde for the thermodynamic profile, dD comes from satellite data from 2005-2009, and d18 comes from the d-excess of the GCM extrapolated to the observed ground-based vapor measurement. There are major concerns with these inputs coming from different data products that all have different spatial/temporal scales and measurement principles.

For example, if the thermodynamic profile and the dH profile were both taken from satellite retrievals, then that dataset would at least be internally consistent. But, taking the different atmospheric properties from different sources, there is little physical basis to believe that they would be internally related to each other.

In my view, there are large sources of error being introduced with each model input assumption, starting with the droplet generation height, continuing with the droplet sizes, and compounding with each additional data product incorporated. I feel these uncertainties are too severe to enable a useful comparison with the in-situ rainwater measurements. Likewise, the measurements are unable to offer an advantage in assessing the model

performance. For these reasons, this reviewer does not see added value in presenting the BCIM in its current form alongside the in-situ rainwater measurements.

**L 426-427**

Clarification on the choice of droplet formation height: I would agree that raindrops are most likely to be generated in regions where CLWC is maximum, but Section 2.4 indicates that the BCIM releases the droplets from cloud base, and it seems implied that this height is what is input to the model. Is the droplet released from the cloud base, or from a height inside the cloud? Showing the input profiles for temperature and RH, as mentioned above would clarify this for the reader.

If the droplet is released from cloud base:

The moist adiabatic lapse rate in the Rayleigh method should provide a cloud base height based on the height where the moist parcel reaches saturation, would it not? Could that be used in place of this fixed 850 mb level?

If radiosondes from the study period are provided, why not use the lifting condensation level (LCL) from the soundings to define the cloud base height?

If the droplet is released from a height inside the cloud:

The authors acknowledge that assuming the same droplet formation height for each day is going to be erroneous. How does this fixed 850 mb height compare with the cloud base height from the moist adiabatic assumption and the cloud base height from the radiosondes on each day? Can the authors confirm that it is above the cloud base in all cases? The ERA5 data in Figure S3 indicates that the cloud base is not often above 850 mb, which is encouraging, but it does vary. Perhaps the authors could instead calculate the average height above cloud base where CLWC peaks, allowing the droplet formation height to be adjusted based on the cloud base height on each day.

**L 431-437**

A citation is needed for the Marshall-Palmer relationship. Looking at the Marshall & Palmer 1948 paper, it may not be an ideal choice for the droplet diameter estimate. Have the authors considered using droplet size distributions from Murali Krishna et. al. 2021, which is included in the references? It appears that Murali Krishna et. al. 2021 has DSD's for a range of rain rates in the Western Ghats which would seem more appropriate than those collected in Ottawa in the Marshall & Palmer paper, and of higher quality using modern measurement techniques.

**L 443-448**

Can the authors please describe or reference the equations used to calculate these profiles.

**Line 470-471**

The authors should provide a brief description and references for the isotope enabled LMDZ GCM model.

Referencing a personal communication to obtain the isotopic inputs to the BCIM is a bit peculiar. If this particular GCM output has not been published prior, and was provided for use and publication in this analysis, this reviewer wonders if it would be more appropriate to include the contributor of the GCM isotope profiles as a co-author.

**Line 522, 556-559**

I find the assertion that "the main source of error in Run 1 and Run 2 could be improper vapor isotope values", to be invalid. Considering the uncertainties around the choice of droplet introduction height, droplet size, and thermodynamic profiles from the radiosondes, there are many possible sources for the lack of agreement with the observations. It is very likely a combination of many sources. The conclusions of L 556-559 are not justified, for these same reasons.

**Figure S4**

The vapor  $\delta D$  profiles in panel (b) of Figure S4 show indications of cloud base heights where the profile transitions from being constant, to undergoing rayleigh distillation within the cloud. This further begs the question of whether or not the droplet is intended to be released at cloud base or not. In many of these profiles it appears that the droplet formation height of 850 mb is quite far above the cloud base.

This also raises questions about the isotopic profiles used in Runs 2-4 that needs to be discussed: Why are the ambient vapor isotopes below the cloud base decreasing with height (and where is the cloud base)? I don't see a physical basis to justify this given that there are strong/dominant surface moisture fluxes, and the expectation of a well-mixed boundary layer.

**Figure S7**

I'm not seeing where the dexc-low line is in the plot?

**Overall organization:**

The manuscript needs significant reorganization for readability and to meet expected standards. Reorganizing the sections could also result in a much more concise paper. The line of logic leading to each conclusion is rather circuitous (not direct). In my opinion the lack of organization makes for a laborious reading experience.

A few opportunities for re-organization are:

- Lines 263 275 is more appropriate for the discussion section, rather than results.
- All of Section 4.3 belongs within the methods and results section.
- Lines 667 676 belong in the discussion section, not in conclusions.

**Line by line comments:**

Check with editor: Should "micro-physics" be hyphenated?

L 78: capitalization of "Estimate"

L 81-82: I think the assessment of previous studies being "often innacurate" is too strong. Suggest for example: "However, it remains a challenge to account for all cloud microphysical processes and their associated isotopic fractionations." The sentence could simply be removed as well.

L 97: "riming" mispelled

L 109: As the referenced study has four years of 1-min disdrometer data, I think the sentence stating "limited to scanty observations" is too strong. Furthermore in the following sentence, I think it is again too strong to claim the current study as "accurate and simpler". I would suggest avoiding these types of qualifier words. For example, the authors could revise to: "The question arises of whether one can determine the raindrop evaporation and its variation using measurements of isotope ratios in rain and vapor".

L 147: The link to this document does not appear to be working. Perhaps the authors can briefly summarize the design and requirements of the rain collection in case the reader is unable to access the document.

L 150: "indigenously" is not the correct word here. Did the authors mean "in-house"?

L 152: units "min-1" needs a superscript.

L 351: punctuation.

L 566: What is "point #4"?

---

## Author Comment (AC1)

**Review #1**

This work evaluates rain and water vapor isotope ratios collected during the monsoon period near Pune, India using a Below Cloud Interaction Model (BCIM). By tuning the boundary conditions of the model, the work concludes that 23% of the raindrop mass evaporates on average near Pune. The work broadens our observational constraints on rain evaporation, which is an important process that influences model climate sensitivity and storm organization and intensity. However, some of the methodology would benefit from additional clarification. Comments to that effect are detailed below.

We sincerely thank the reviewer for appreciating our work, evaluating the manuscript critically, and providing many constructive/useful comments and suggestions. Accordingly, we have revised the manuscript (RM1 hereafter). The responses to specific queries are given below in blue font.

1. BCIM input, set up, assumptions, uncertainties:

To run the BCIM requires an assumption about the background water vapor isotopic profile. The paper tries using a) a Rayleigh distillation, b) a profile from GCM output, c) an average satellite profile from TES, and d) tuning the isotope ratios so that the BCIM rain and vapor isotope ratios match observation. Only the last attempt produces an agreeable result, largely because it was specifically tuned to do so. Should we be concerned that assumptions about other factors—drop size distributions, for example— could actually be causing the model-observation discrepancy and that their effects are simply being masked in the tuning?

R-1-1: The ambient vapour isotope values have the maximum impact on the model rain isotope values. This can be shown by multiple regression. We carried out a multiple regression analysis of rain isotope values with four influencing factors (RH, Temperature, surface $\delta Dv$ and drop diameter) in their normalised forms. The normalised values of the model rain isotope ratios $\delta D_{mod\text{-}rain}$ for the 29 days (selected for BCIM runs) were regressed with the four aforementioned variables. We obtain the following multiple regression equation

$\delta D_{mod\text{-}rain}$= -0.114*RH+0.035*Temperature-0.059*diameter+0.986*$\delta Dv$

The above equation indicates that the major influence on the model rain isotope value is from the ambient vapour $\delta Dv$ (with a coefficient of nearly one meaning 1% change in $\delta Dv$ would result in ~1% change in the rain $\delta Dv$). In contrast, the influence of RH, for example, is only one-tenth for the same percentage change. Therefore, for tuning we changed the vapour isotope value only and no other factor. We have included this discussion in the RM1.

Some of the answer appears in the Supplemental, and I strongly suggest that this material be included in the main manuscript instead.

R-1-2 The supplementary part is included in the RM1 now.

In fact, Figure S7 suggests that results vary more strongly with either drop size or RH than with assumptions about the background isotopic profile. More in depth discussion

of assumptions and uncertainties would be helpful—particularly uncertainties in the rain evaporation percentage, as well as clarification on BCIM input methodology.

R-1-3: We would like to clarify here that the influence of various input parameters on the model rain isotope values is different from their influence on the raindrop evaporation. While input vapour isotope profile mostly control rain isotope values (see reply R-1-1), the drop size has the maximum impact on the evaporation fraction (see reply R-1-14). In fact, background vapour profile has negligible influence on the evaporation which is determined by the cloud microphysics involving only the three other factors. A multiple regression analysis using normalised values of evaporation fraction regressed with drop diameter, RH and temperature (see R-1-14) show this clearly. As suggested, more detail discussions on the BCIM input values and sources are given now.

Table R-1-1: Input parameters for various BCIM runs

| Number | BCIM input parameter | Data source for Run-1 | Data source for Run-2 | Data source for Run-3 | Data source for Run-4 |
|---|---|---|---|---|---|
| 1 | Drop size | Marshal-Palmer equation using rain rate from IMD | Marshal-Palmer equation using rain rate from IMD | Marshal-Palmer equation using rain rate from IMD | Marshal-Palmer equation using rain rate from IMD |
| 2 | RH profile | Rayleigh ascent ~15 % increase per km | Average Radiosonde normalized to ground value | Same as Run-2 | Same as Run-2 |
| 3 | Temperature profile | Rayleigh ascent Lapse rate ~ 5.6°C per km | Average Radiosonde normalized to ground value | Same as Run-2 | Same as Run-2 |
| 4 | dDvap profile | Rayleigh ascent ~7 ‰ decrease per km | TES normalized to measured ground value | $\delta D$ values reduced slightly to maintain the shapes similar to Run-2 | Nearly the samew profile as Run-3 |
| 5 | dexvap profile | Rayleigh ascent ~0.1 ‰ increase per km | LMDZ $\delta D$ and $\delta^{18}O$ values used to get dex vap normalized to measured ground value | d-exc vap decreased from Run-2 average of 17‰ (average ~7‰). | d-exc vap decreased from Run-2 average of 17‰ (average ~10‰). |
| 6 | Rain drop introduction height | ERA-5 Cloud Liquid Water Content peak | ERA-5 Cloud Liquid Water Content peak | ERA-5 Cloud Liquid Water Content peak | ERA-5 Cloud Liquid Water Content peak |
| 7 | Cloud base Height | LCL | LCL | LCL | LCL |

For example,

- I'm unsure why TES from 2005-2007 was averaged over a 4x4 degree grid box to represent the profile at a single observation site in the year 2019. AIRS or IASI would have provided 2019 data with much greater frequency and thus created an opportunity to evaluate not just a mean profile but variations from that mean and their impacts on model output.

R-1-4: As per the reviewer's comment and suggestion, we have explored the AIRS (over 1° x 1° grid centred around our study location) and IASI datasets. We found that IASI datasets are not available for our study period (June-September 2019). We did two specific exercises with the AIRS datasets which are available for the period. The

constructed profiles for the selected days are shown in Fig. R1-1 along with the TES profiles that were used.

[Figure]

Figure R-1-1: (a) Vertical variation of vapour $\delta D$ values as obtained from AIRS dataset (orange envelope) and the $\delta D$ profiles constructed from TES values used for Run-2 (blue envelope). (b) same as (a) only the surface values from AIRS dataset normalized to our surface observation (c) scatter plot showing model and observed rain $\delta D$ values after Run-2 using TES $\delta D$ adopted profiles (red filled squares) and AIRS $\delta D$ profiles with (green filled circles) and without surface normalisation (blue filled circle). The run was performed for three specific days- one with maximum observed rain $\delta D$ values, one for minimum and one close to mean vapour $\delta D$ values. (d) same as (c), but for $\delta^{18}O$ values.

As one can see, the values are depleted at the surface level compared to our measured values (Fig. R1-1a). We understand that this discrepancy stems from the less accuracy of the satellite data in the near-surface or boundary layer. Therefore, we increased the mean extrapolated AIRS surface value linearly to match our surface observations (Fig. R1-1b). As AIRS data do not provide $\delta^{18}O$ values, we computed $\delta^{18}O$ values from $\delta D$ obtained from AIRS and d-excess profiles used in Run-2. We compared observed and model $\delta^{18}O$ (Fig. R1-1c) and $\delta D$ (Fig. R1-1d) values. As evident from these figures, the model $\delta D$ and $\delta^{18}O$ values are far off from the 1:1 line compared to model calculated on the basis of TES profile. This clearly demonstrates that using the AIRS profiles, we do not get any improvement and in fact, there is further disagreement with the measured values.

- I'm also concerned that multi-order polynomials are used to interpolate the vertical profiles, when there are not a sufficient number of observations to constrain the number of coefficients.

R-1-5: We have discussed explicitly the limitation of our choices of vapour isotope profiles in L684-692. The higher order gave better agreement with the boundary values for the $\delta$Dvap and d-exvap. We explain the adopted procedures below to clarify the issues involved in the adoption of the profiles.

The input profiles used in the BCIM are associated with a number of limitations:

- We used TES satellite data averaged over 2005-2009 to guide our choice of vapour isotope profiles, but the year of analysis was 2019. In the absence of any other data, there is no way to ascertain the degree of deviation of the true profile of each sampling date from the adopted ones in Run-2 except boundary constraints (observed ground values).
- The d-excess profiles were adopted based on the $\delta$D and $\delta^{18}$O profiles obtained from the LMDZ model. As discussed in the main text, individually the two isotope ($\delta$D and $\delta^{18}$O) profiles from the LMDZ did not give us good agreement with the observations but after little tuning the d-excess gave good agreement with the observed rain values.

- The isotope profiles were constructed using ground observations as boundary values. Even then, there was slight mismatch with the observed values. We had to tune $\delta$D, $\delta^{18}$O values and d-excess values of vapour moderately to achieve good agreement.

Below we explain in detail how we constructed the $\delta$D and d-excess profiles for each date.

1: We analyzed available TES $\delta$D profiles (digital values) for the years 2005-2007 and adopted three profiles which correspond to the Minimum, Mean and Maximum surface vapour isotope values of -125.2‰ ($v_1$), -80.8‰ ($v_2$) and –63.0‰ ($v_3$).

2. These three profiles were fitted with equations of the type: $Ah^3+Bh^2+Ch+D$ where h is the altitude giving us three coefficient sets of A, B, C and D values for each of the surface $\delta$Dv values ($v_1$, $v_2$, $v_3$) given above.

3. We now need to find the coefficient sets for each day where the surface values change. This was done by fitting each coefficient as a function of the surface value (v) using data from 2 (above). Once we got the fitting equation ($av^2+bv+c$) for each coefficient (A or B or C or D) as a function of the surface value v, we can get the coefficient for each day corresponding to the surface value (v) observed for that day.

This exercise was necessary to translate the digital TES values into an analytical form, allowing for the easy calculation of vapour isotope values at each height (at one meter resolution required for the BCIM inputs) from the uppermost drop introduction point to the ground level, resulting in a smooth shaped profile.

A similar exercise was conducted to obtain the daily d-excess profile from the LMDZ GCM output for Pune in 2019 and normalising the profile to the surface-measured vapour d-excess value. In brief, this was done by using the available $\delta$D and $\delta^{18}$O profiles from LMDZ output for three cases (Mean, Max and Min surface values), fitting the appropriate polynomials and then from these constructing the d-excess profiles for three cases with five coefficients. Again, fitting was done for each of the polynomial

coefficients (A, B, C, D and E) as a function of surface value and then use them to get the d-exc profile for each day.

This means that by using the observed ground vapour value as a boundary value we obtain the desired profiles for $\delta D$ and d-exc from an analytical fitting of digital data from TES (adopted) and LMDZ. We used a multi-order polynomial to get the fitting as accurately as possible, especially for d-excess, because while estimating uncertainty, we can consider the uncertainties in the TES and LMDZ without worrying about the fitting uncertainties. We show the technique for a three-parameter case for $\delta D$(vapour) profile initially used for Pune. The fit has three coefficients A, B, and C (later we used four coefficients).

[Figure]

Fig R-1-2: (a) vapour $\delta D$ (‰) profile as a function of altitude (mbar). The three profiles were constructed on the basis of mean value, maximum value and minimum value at the surface available from the TES data set that were downloaded. The second order polynomial fit ($y=Ax^2+Bx+C$) used to fit the digital data as a function of altitude (for each case) are also given in the figure. The coefficients A, B and C as a function of surface vapour $\delta D$ values are plotted in (b), (c) and (d) respectively and fitted equations are derived. These equations can now be used to derive the coefficient set for any given date using the measured surface value for that date.

We note that the three TES profiles are nearly parallel and change only little within 950 to 850 mbar which was the relevant interval for below cloud zone over Pune. If we take the day-to-day changes in the CLWC peak considered to be the drop introduction height- we need to go up in these profiles to get the vapour $\delta D$. A, B and C are the surface value dependent coefficients – these are fitted with surface value as

variables (shown above). Once we have the coefficient set for each date, we can use the fitted polynomials for calculating vapour $\delta D$ as a function of height for each day.

Other places to clarify:

L 435 says, "drop diameter at the ground is provided as input," but how does this work? Isn't the drop size at altitude the initial input for the BCIM?

R-1-6: Drop size at the introduction altitude is not given as input. The drop diameter at the surface is given as an input. Then the BCIM estimates the drop diameter at the height of introduction through iterations. To do this iteration, only the microphysics part dealing with the major isotopomer $H_2O$ (based on temperature and RH profile) is needed (Graf et al 2019). Typically, if the terminal diameter at the surface is 1 mm, the diameter at the introduction is about 1.25 mm. We have partly revised the text as below:

*"The drop diameter at the ground is next provided as an input, and then the initial size at the drop height (about 1.5 km above ground) is estimated iteratively in BCIM using the microphysics part of the model dealing with the major isotope $H_2O$ (based on temperature and RH profile; Graf et al., 2019)."*

L 469 says, "appropriate interpolations were carried out." The interpolation method should be specified.

R-1-7 We have added the following discussion in the RM1 – *"The relative humidity and temperature data were obtained from radiosonde observations (twice a day; 00:00 ZZ and 12:00 ZZ) at IMD observatory in Pune. Radiosonde observations are available around every 50mb pressure interval. On the other hand, the input is required at 1-meter intervals. Therefore, a linear interpolation between every two consecutive pressure levels in logarithmic scale (Ingleby et al 2014) is carried out to obtain RH and T values at each intermediate level. As, the BCIM requires RH=100% for formation of water droplets (Graf et al, 2019), the RH values above lifting condensation level (LCL) were considered as 100% The above sentence is added in the RM1"*

L 479: "The procedure is discussed…" but where? Below?

R-1-8: We are sorry for this inadvertent mistake. Yes, the procedure is discussed below.

L 500 says the constants were estimated "by interpolation". What kind? Linear? More detail would be helpful.

R-1-9: Please see the detail discussion given in R-1-5. This was done by fitting a second-order polynomial to each coefficient (A, B, C, D and E) and then using that polynomial to estimate the coefficients corresponding to the ground vapour value on a given day. This has been explained in the revised text as below *"The polynomial coefficients (four in number for $\delta D$ and five in number for d-excess) were calculated for three cases: (1) for the maximum observed surface vapour D/H value, (2) for the mean surface value, and (3) for the minimum observed value, giving us three sets of A, B, C, D and E values. The constants for each day (with different values of $\delta D$ vapour and d-excess at ground as measured) were next estimated by linear interpolation using these three sets. Obviously, this method of*

*interpolation constrained by surface vapour measurements assumes that the vapour aloft is related to the surface value."*

L 540 talks about "achieving a reasonable agreement" through tuning. But more specifics are needed. What makes something reasonable? Is there a physical basis?

R-1-10: This is a good point. We carried out a two-tailed Student's t-test with observed and modelled $\delta^{18}O$, $\delta D$, and d-excess values after each run. The null hypothesis is one where the observed and model population means are not significantly different, and we chose our significance level $p=0.05$. We look for that run when p is more than the significance level for all three isotope ratios. The p-values for all four runs are given below.

*Table R-1-2: The p-values of student t tests performed with observed and model rain isotope ratios ($\delta^{18}O$, $\delta D$ and d-excess)*

|  | Run-1 | Run-2 | Run-3 | Run-4 |
|---|---|---|---|---|
| $\delta^{18}O$ | 0.54 | 0.34 | 0.57 | 0.86 |
| $\delta D$ | 0.45 | 0.44 | 0.84 | 0.84 |
| d-excess | 0.00 | 0.00 | 0.02 | 0.95 |

We consider a reasonable agreement when the null hypothesis is valid for all three parameters ($\delta^{18}O$, $\delta D$, and rain d-excess). p-values in in Run-1, Run-2 and Run-3 do not qualify (p is less than 0.05 for d-excess) but Run-4 meets this criterion (p values for all three are greater than 0.05). Therefore, the null hypothesis is valid in Run-4 (the model means are not significantly different from the observed means). We have added this discussion in the revised manuscript, Section 4.3.

Although we chose a statistical criterion above for reasonable agreement, it also has a physical basis. The physical basis behind a reasonable agreement would be the validity of the assumptions of the cloud microphysics of the BCIM, which calculates the evolution of the raindrops as they fall through the cloud and below-cloud layers. The evolution changes the size as well as the isotope ratio, finally culminating in the values that we measure at the ground. We claim that a reasonable agreement of the isotope values is proof that the assumed cloud microphysics of the BCIM (with all assumed input parameter values) holds good in the case of Pune rains, and therefore the drop evaporation estimated from the same model can be considered valid.

L 557-8 talks about needing to increase the d18O and decrease the dD profiles, but how is this done? Uniformly with height? (This seems to be what the figure shows). Or only near the surface or top? Again, further clarification would help.

R-1-11: The constructed isotope profiles are not uniform with height if we consider a much higher altitude than ~800 mbar, say about 400 mbar. The isotope values ($\delta^{18}O$ and $\delta D$) following our polynomial equation decrease with altitude and profiles for various days converges at about 325 mbar. See the figure in reply R-1-5.

In fig. 4c, we have shown the isotope profiles only up to the height of our interest (that is the drop formation altitude). The required tuning (for Run-3 and Run-4) was indeed such that the changes were nearly the same with height (about 1 km as shown in the figures) thus keeping the profiles nearly parallel to the ones assumed before the change. The profiles are shown in the SI; they vary only slightly for the sub-cloud layer (within about 5-10 per mil). Therefore, one can indeed say that the changes on the profiles are nearly the same in this height range. For comparison, please see the SI plots of the pre and post tuning profiles. The reviewer may also check the adopted TES profiles between the levels 950 and 800 mbar in the figure given in reply R-1-5 showing how they are nearly parallel.

L 525 says "one possible explanation [for error in Runs 1 and 2]" might be due to a missed ET signature.

R-1-12: The possible role of ET has been discussed in our manuscript (see below). However, we discounted that possibility based on a few earlier studies.

***"One possible explanation could be a significant contribution from the small-scale local surface moisture having a different isotopic composition (evaporation or evapotranspiration from water bodies or trees within a few hundred meters). However, this possibility can be ruled out as a study using satellite data showed that due to high humidity and low temperature during ISM, evaporation/ evapotranspiration (~0.5 mm day$^{-1}$) adds a negligible amount of moisture compared to the advective fluxes in this region (Pathak et al., 2014)."***

But what about the fact that the Runs are using average profiles from a GCM and 4x4 degree satellite average as a proxy for a single convective location?

R-1-13: This is an important issue. The reviewer has pointed out the usage of two datasets as input of the BCIM. While a mean $\delta D$ profile averaged over a 4°x4° area from TES has been used to obtain the shape of the altitudinal variation, the daily vapour $\delta D$ profiles for the model input were constrained to pass through the measured surface values (keeping the same shape). The daily output of LMDZ (a GCM) is used to obtain the daily d-excess profiles. We address the issues related to choice of vapour isotope profiles below.

Firstly, the BCIM is based on the input vapour profile up to the level of the drop introduction. This is because BCIM assumes a single column of vapour over the location where sub-cloud processes operate as the drop falls. This is undoubtedly a simplified picture of the real situation and has been discussed by the proponents of this model (Graf et al., 2019). In the case of monsoon rains, no measurement of vapour isotope profile over Pune (or any place in India) has ever been done. One possible way is to provide the surface measurements of the vapour isotope and other parameters to BCIM and allow the model to estimate their profiles considering a Rayleigh adiabatic ascent (Run-1 of Fig. 6). Although, we obtain some correlations between observed and model rain isotope values, the results were not satisfactory (see reply R-1-10 and Table R-1-2 for a statistical criteria). Therefore, we had to construct the isotope profiles for each day using other available data sets like TES and LMDZ.

Secondly, we admit that using the TES data for a bigger domain (4°x4°) is a limitation of this study, which has also been mentioned when we discussed the sources

of uncertainty. However, we note that we did not just use the TES average profile but normalised it to the ground-measured value of each day. We maintained the shape but shifted the profile to constrain it to the measured boundary value (please see reply R-1-5 for the detailed procedure). Another alternative approach could have been to utilise satellite data from AIRS (for vapour $\delta$D input profiles), which is available during our study period and offers better temporal and spatial resolution than TES. We followed this suggestion by the reviewer, but the results were not satisfactory (see reply R-1-4). This makes the procedure we followed to obtain vapour $\delta$D profiles the only viable option.

Thirdly, none of these satellite data provides vapour $\delta^{18}$O values but the BCIM needs vapour d-excess (that is both $\delta$D and $\delta^{18}$O) as an input. Now, in the absence of any other source for this profile, we used the LMDZ GCM + TES combination, which provides vapour isotope profiles (both $\delta^{18}$O and $\delta$D) over our region at coarse resolution. The only justification we have for this is that this assumption yields the rain values close to those observed, though not quite accurately (model rain drop $\delta^{18}$O is slightly lower, on average, by 0.4‰ in $\delta^{18}$O). Correspondingly, the model is higher in most cases for d-excess (by 0 to 15‰), on average by 7‰ (Run-4). Therefore, we needed a slight adjustment of $\delta^{18}$O (increase the values) and $\delta$D (decrease the values) to match the values for all three ratios: $\delta$D, $\delta^{18}$O and d-excess. The increase in $\delta^{18}$O and decrease in $\delta$D together decrease the d-excess of the vapour.

This tuning is small and can be justified considering the uncertainty in the assumptions (local validity of LMDZ+TES) as pointed out by the reviewer. It is clear that the average TES profile used (even though constrained by the boundary values) was not accurate enough and needed slight adjustment. Before choosing TES, we tried out LMDZ and ISOGSM but did not achieve a close agreement like TES (results not shown).

2. RH as the primary control:

The paper argues that RH is the primary control on raindrop evaporation because RH varies more than other factors like drop size diameter. But I worry that this is inferred from an absolute value comparison when what might be more relevant is to evaluate how the standardized values differ. One can do this by standardizing the predictors or by looking at partial coefficients of determination in the regression. Without that additional step, I'm not sure that this argument is well supported. Moreover, earlier in the work (e.g. L 392) there seems to be a stronger emphasis on the importance of drop size. Is there a reason that the argument (apparently) shifts?

R-1-14: We are thankful to the reviewer for the suggestion to consider normalized values. We have removed the earlier discussion in terms of absolute values and added the following discussion in terms of normalised values.
We carried out the multiple regression analysis with the standardised values of the parameters influencing raindrop evaporation fraction. The standardised value converts the parameters to comparable relative scale (unit independent); it is calculated by taking the value of any parameter for any specific day minus the mean value and dividing by 1$\sigma$ standard deviation. This was done for all days. The following multiple regression was obtained,

Evaporation Fraction = -0.329*RH +0.370* Temperature -0.665* Diameter

As standardised values of all parameters are used, the values have no units. As evident by the coefficients, diameter is the most dominant factor controlling the evaporation fraction. The above equation shows that a 10% increase in drop diameter can reduce the drop evaporation fraction by ~7% while a similar change in RH (temperature) can decrease (increase) the evaporation fraction by ~3% (4%). The reviewer is right that RH is not the dominant factor but drop diameter is.

We have revised the discussion, conclusions and other parts of the manuscript accordingly.

3. Other minor comments:

L 237 talks about 0.5 standard deviations: this is an unusual choice. Why not go with a more standard 1- or 2-sigma envelope?

R-1-15: We agree and have modified the part as *"The mean and 1σ standard deviation of $\delta^{18}O$ and d-excess values of rainwater are −1.3±2.6 ‰ and 3.9±2.7 ‰, while those of the vapour are -12.5±2.5 ‰ and 18.3 ± 5.2 ‰, respectively."*

Figure 2 talks about four regions marked by shading, but they are actually bounded by vertical lines.

R-1-16: Thanks for pointing this out. The caption of Figure 2 has been revised: *'The four grey vertical boxes (numbered 1, 2, 3, and 4) denote synchronous low OLR values'.*

L 296 talks about d-excess increasing while d18O decreases. The evaporation actually causes the rain d-excess to decrease and the d18O to increase. So while the relationship is self-consistent, it might be switched for clarity.

R-1-17: We agree. The suggested change has been made as *"This is seen in our study (Fig. 3b) where rain d-excess decreases with an increase in $\delta^{18}O$ values."*

L 322 talks about deep convective systems being "controlled by different microphysical processes." But, different from what?

R-1-18: Thanks for suggesting a clarification. "different" is a wrong word to use. The sentence is revised as follows:

*"Risi et al. (2023) have noted that in the tropics, most of the precipitation falls under deep convective systems, which are controlled by various microphysical processes (like rain evaporation, diffusive liquid-vapour exchanges, and mesoscale downdrafts) connected through mesoscale circulations."*

L 375 provides a specific range, but this is only approximate. No sample actually gets to -20 ‰. Also, the word "respectively" is not needed after the parenthesis.

R-1-19: Thanks for pointing this out. The correct ranges are now added*. "The time series of Δδ values show (Fig. 5a) that for Pune the precipitation samples varied between -15 ‰ and 21 ‰. For Δd, the time series shows negative values in all cases (ranging from -2 to -24 ‰)."*

The word "respectively" is deleted.

Figure 5 might benefit from marking the 15 "evaporation" samples so that one can pick them out more easily.

R-1-20: We agree. The 15 "evaporation" samples are marked by serial numbers in revised Fig. 5.   See below-

[Figure]

*Figure R-1-3. (a) Time series of Δδ and Δd of the rain samples collected during 2019 monsoon (July to September) in Pune. Δδ and Δd values (total points=29) as defined in the text following Graf et al. (2019). The dotted line indicates Δδ=0. All data points where Δδ>0 are marked with numbers totalling 15. (b) Time series of daily average surface temperature and relative humidity recorded at IMD Pune observatory during the study period.*

It would also help to clarify somewhere in the text that Figure 4 shows absolute value differences between rain and vapor while Figures 6 and 7 show the difference when the rain values are converted to vapor in equilibrium with the rain. It took me a while to understand why I was seeing different axes values.

R-1-21: Thanks for pointing this out. We have accordingly modified the symbol: from Δd-excess (w-v) to Δd-excess$_{r-v}$ and from $\Delta\delta^{18}O$(r-v) to $\Delta\delta^{18}O_{r-v}$ in Figure 4 and its caption. The subscript r-v denote rain relative to vapour. For clarity, we defined the term in the text. "*The current study exhibits a reasonable anti-correlation between the absolute value differences in d-excess (Δd-excess$_{r-v}$) and $\delta^{18}O$ ($\Delta\delta^{18}O_{r-v}$) of rain and vapour (Fig. 4b).*" The revised figure is shown below-

[Figure]

On the other hand, the Δδ and Δd referred to the difference between the surface vapour values and the vapour in equilibrium with the rain. This is clarified in the revised manuscript "***The difference between isotopes (δD and d-excess) of vapour in equilibrium with raindrops and the observed vapour (at the ground level, defined as Δδ and Δd, respectively) is useful to quantify the departure from equilibrium.***"

L 630: "intimate relations" is not the right phrase for this context. Correlations?

R-1-22: The line has been revised as 'The association of these events with the regional meteorological processes is studied further.'

Conclusion #4: I find this point confusing, but maybe I just need to reflect on it a bit more. It suggests that local water vapor supply cannot be important to sub-cloud moisture, even though the paper argues that rain evaporation fraction is significant. Is

the issue that a lot of the raindrop mass is evaporated, but that total amount of water vapor yielded is actually quite small?

R-1-23: The reviewer has understood correctly. The contribution to the vapour mass overhead is small from the evaporation. Typically, about 5% of the vapour is converted to rain and 25% of the rain evaporates. This makes the net contribution of rain evaporation-induced vapour to the overhead vapour (a large reservoir) small, approximately 1 to 2%. However, the effect on the rain mass falling on the ground (smaller reservoir) is significant, averaging about 25%. This is expected from the isotope mass balance, where the effect on the smaller reservoir is greater than that on the larger reservoir.

Collision-coalescence is not mentioned even though it is an important precipitation process that the BCIM neglects.

R-1-24: We agree with the reviewer and the following part has been added to section 5 (Conclusion and limitations)

*"Although some studies pointed out that collision-coalescence is an important warm rain process that occurs in various rain shadow regions of India (Padmakumari et al., 2024) including those in WG (Konwar et al., 2014), BCIM neglects this process while estimating rain isotope values. We admit this may introduce an error in rain evaporation estimation which likely could be small but cannot be quantified in the present study."*

References

1. Graf, P., Wernli, H., Pfahl, S., and Sodemann, H.: A new interpretative framework for below-cloud effects on stable water isotopes in vapour and rain, Atmos. Chem. Phys., 19, 747–765, https://doi.org/10.5194/acp-19-747-2019, 2019

2. Ingleby, B., Pauley, P., Kats, A., Ator, J., Keyser, D., Doerenbecher, A., Fucile, E., Hasegawa, J., Toyoda, E., Kleinert, T., Qu, W., James, J.S., Tennant, W., and Weedon, R.: Progress toward High-Resolution, Real-Time Radiosonde Reports, Bulletin of the American Meteorological Society, 97 (11), 2149–2161, https://doi.org/10.1175/BAMS-D-15-00169.1, 2016

3. Konwar, M., Das, S.K., Deshpande, S. M., Chakravarty, K., and Goswami, B. N.: Microphysics of clouds and rain over the Western Ghat, J. Geophys. Res.-Atmos., 119, 6140–6159, https://doi.org/10.1002/2014JD021606, 2014.

4. Padmakumari, B., Maheskumar, R. S., Morwal, S. B., and Kulkarni, J. R.: Variability of Index of Coalescence Activity (ICA) over a rain-shadow region during monsoon and its role in cloud seeding programs in India. Atmospheric Research, 304, 107390. https://doi.org/10.1016/j.atmosres.2024.107390, 2024

5. Pathak, A., Ghosh, S., and Kumar, P.: Precipitation Recycling in the Indian Subcontinent during Summer Monsoon, J. Hydrometeorol., 15, 2050–2066, https://doi.org/10.1175/JHM-D-13-0172.1, 2014

6. Risi, C., Muller, C., Vimeux, F., Blossey, P., Védeau, G., Dufaux, C., and Abramian, S.: What Controls the Mesoscale Variations in Water Isotopic Composition Within Tropical Cyclones and Squall Lines? Cloud Resolving Model Simulations in Radiative-Convective Equilibrium, J. Adv. Model Earth Syst., 15, e2022MS003331, https://doi.org/10.1029/2022MS003331, 2023.

**Review #2**

**"Assessing raindrop evolution over northern Western Ghat from stable isotope signature of rain and vapour"**

**Summary:**

The investigators aim to assess the degree of evaporation of rain between cloud base and ground level by linking ground based water isotope measurements with a below cloud isotopic model.

Ground based samples of rainwater and ambient vapor, located at the Indian Institute of Tropical Meteorology in Pune, are collected throughout the 2019 summer monsoon season. It appears that rainwater was collected on a 24 hr basis. Vapor was collected by pumping air through a chilled vessel to condense moisture out of the air, with a time resolution of 3-4 hours. Vapor and rainwater were sometimes collected on the same day, and are collocated.

The water isotopic trend throughout the season is analyzed in the context of large scale meteorology. Using outgoing longwave radiation anomaly (from satellite), a link between mesoscale convection and depletion of rainwater isotopes is demonstrated. Analysis of observed rainwater and vapor isotopes in $\Delta\delta$-$\Delta$d space indicates the prevalence of below-cloud evaporation taking place to various extents throughout the season. Additionally, there are signatures in the vapor isotope data which lead the authors to believe that depleted vapor is being downdrafted potentially as a result of raindrop evaporation.

The authors then attempt to use the BCIM model to compare modeled raindrop isotopic evolution with the ground based measurements. Several estimates must be provided as inputs into the model: droplet formation height, droplet size, atmospheric temperature, humidity, and vapor isotopic profiles. The authors describe three approaches:

1. Using ground based temperature, relative humidity, and vapor isotopes, assume a moist adiabatic atmospheric lapse rate. The droplet begins at a fixed height of 850 mb. Droplet sizes are calculated from daily rainfall rates.

2. Radiosondes are used for thermodynamic profiles. Vapor dD is obtained from 2005-2009 satellite retrieval, while dO18 is obtained using d-excess profiles from a GCM and extrapolating from the observed vapor d-excess. Droplet sizes from daily rainfall rates. 850 mb droplet formation height.

3. Vapor isotopic profiles from method 2 are tuned to produce a d-excess at ground-level value in agreement with the observations.

The authors use the final tuned model input to conclude that on average 23% of rainfall during the season evaporated during descent.

We sincerely thank the reviewer for evaluating the manuscript critically and providing many constructive/useful comments and suggestions. Accordingly, we have amended the manuscript parts of which are reproduced here for clarity. The replies to the general and specific comments are given below in blue font.

**Overall assessment:**

This study presents a new set of collocated rain and vapor isotope observations, a measurement approach which holds great potential for addressing the topic of interest here. The simultaneous observations of vapour and precipitation (as opposed to only one or the other) is particularly advantageous.

**R-2-1:** We thank the reviewer for appreciating our study using rain and vapour isotope analysis in Pune, India.

However, with the exploration of the BCIM that is presented here, there is too much going on in this study, making the results muddy. This reviewer believes that the impact of and insights from the in-situ measurements is detracted by the troubled attempt to reproduce the observations with the BCIM.

**R-2-2:** We agree with the reviewer's comment about the lack of clarity in the way the BCIM model is presented in the manuscript. We have now revised the presentation by incorporating comments from the present reviewer as well as the reviewer #1 who also pointed out some deficiencies. Our sincere thanks to both of them.

This reviewer does not find the methods used to initialize the BCIM model to be convincing and therefore holds little confidence in the quantitative values for evaporated fraction obtained from the model. While acknowledging and commending the work that the authors have devoted to the model analysis, this reviewer feels that in its current state it can be omitted entirely.

**R-2-3:** This issue must be addressed in detail, as it deals with the main objective of our study.

First of all, it should be emphasized that the study does not neglect to present whatever can be learnt from pure data analysis. We have indeed discussed the observational aspect to derive important conclusions, albeit qualitatively. For example, we show that there are 4 low isotope events which are correlated with OLR anomaly (indicative of intense convective episodes). The LMWL is characterized by lower slope and there is anti-correlation between rain $\delta^{18}$O and d-excess (Fig 3b); both indicate significant rain drop evaporation. A $\Delta\delta$-$\Delta$d correlation plot shows that none of the rain samples is in equilibrium with the corresponding ground-level vapour; the regression line for $\Delta$d/$\Delta\delta$ has a slope of -0.45 for Pune monsoon rain samples in contrast to the Swiss study (Graf et al., 2019; for rains induced by frontal systems) which show a slope of -0.3. We discussed what this difference indicates in terms of relative magnitude of kinetic fractionation. However, the slope difference cannot be translated into any quantitative estimate of drop evaporation because there are several parameters involved. These points are discussed in the context of earlier studies. All the aforementioned analyses suggest that raindrops in our study area suffer significant evaporation (qualitatively, not quantitatively).

There are many studies from the Indian Monsoon region (including ours) which have demonstrated earlier that drops suffer evaporation during the monsoon but these studies could not derive the extent of evaporation accurately (Midhun et al., 2013; Rahul et al., 2016; Oza et al., 2020; Nimya et al., 2022; Sengupta et al., 2023). Our primary objective in this study was to take one step further and compute the evaporation fractions for each rain episodes over Pune, a rain shadow region. Such a study has recently been done over the Atlantic in a similar way as ours (Sarkar et al., 2023) but has not been attempted for the South Asian Monsoon. Drop evaporation reduces the rainfall and impacts the monsoon heat budget and prediction. This was appreciated by the Reviewer-1 who commented: "The work broadens our observational constraints on rain evaporation, which is an important process that influences model climate sensitivity and storm organization and intensity."

It is clear that to quantify the drop evaporation fraction, we need a model describing the fate of a drop as it falls in the sub-cloud zone. An easy choice could have been to take the available outputs directly from any isotope-enabled GCM that estimates the drop evaporation fraction using its in-built convective parametrization scheme. However, our earlier study (Nimya et al., 2022) has shown that these models

largely underestimate the drop evaporation fraction, which is reflected in the model rain isotope biases. This prompted us to investigate an available one-dimensional model to check if that can quantify the rain evaporation.

We found that BCIM is a simple model which was used in two earlier studies to understand raindrop evaporation and estimate its fraction (Sarkar et al., 2023; Graf et al., 2019). In this model, a single hydrometeor is introduced at a given height whose composition is determined by the equilibrium fractionation from the ambient vapour. During its descent it evolves by incorporating several inputs (vapour isotope profile, RH and temperature profiles, drop size and its introduction altitude and several cloud microphysics parameters). This is termed "initializing" by the reviewer. We had access to some of these parameters on daily scale through real measurement but for others we had to rely on reasonable estimates based on available data sources. We note that despite the difficulties in estimating some of these parameters, we could obtain rain isotope values reasonably close (Run 2) to the measured values. Run 3 and Run 4 were our attempts to match the rain values exactly by tuning the vapour isotope profile, which is known to have the maximum influence on rain isotope ratios (from sensitivity study and please see reply to Reviewer-1, R-1-1). However, for the quantitative determination of the evaporation fraction, we do not need this isotopic tuning. The tuning only helped us to accurately obtain data for the $\Delta\delta D$-$\Delta d$-excess plot and compare the slope with the observational slope. The tuning was minor in magnitude. We agree that these issues should be discussed more clearly. Since the model reproduces the rain isotope values quite well, we surmised that we could use the calculation part dealing with the major isotopomer ($HH^{16}O$) to derive mass loss from the drop as a measure of evaporation. This is taken as an assurance that the cloud microphysics part of the calculation, which determines the evaporation, is reasonable.

Based on the above we would like to continue with the model but take the advice that heavy revision of the model part should be done. As per suggestion, we would change the abstract to stress the observational aspects. We show below briefly how we plan to do this revision.

It is therefore not ideal that the focus of the abstract is based on the BCIM results. A more impactful study might focus thoroughly on the observational dataset,

strengthening the analysis in that area, and removing or heavily revising the modeling component.

**R-2-4:** We broadly agree. As per the suggestion, the abstract has been revised substantially (see below). We have now emphasized the observational aspect and the BCIM part has been thoroughly modified.

*Abstract: We measured rain and vapour isotopes in samples collected simultaneously from Pune, India during the 2019 summer monsoon. The heavy isotopes of both ($\delta^{18}O$ and $\delta D$) were significantly depleted in four events when the Outgoing Longwave Radiation showed strong negative anomaly suggestive of large-scale convection. The $\delta^{18}O$ of the rain samples are negatively correlated with d-excess indicative of modification of rain drops by evaporation. Analysis of the isotope data indicates isotope exchange between rain and ambient vapour and significant raindrop evaporation in the sub-cloud layer. The data plotted in terms of $\Delta\delta D$–$\Delta dex$, where $\Delta$ indicates difference between rain-equilibrated vapour and the ambient surface vapour, show points lying in both 3$^{rd}$ and 4$^{th}$ quadrants, suggesting nearly equal effect of equilibrium exchange with ambient vapour and drop evaporation during the monsoon rain events.*

*We used one-dimensional Below Cloud Interaction Model to quantify sub-cloud processes affecting raindrop evolution in our tropical region. A Rayleigh ascent in BCIM overestimates rain isotope values, although model and observed values are well correlated. Using radiosonde-based temperature and humidity profiles and constructing vapour isotope profiles from a combination of satellite (Tropospheric Emission Spectrometer) data and the LMDZ model outputs, simulations improve and good agreement of model with observed values can be obtained. Sensitivity studies reveal that model outputs are strongly influenced by vapour isotope profiles, and moderately by drop size, temperature and relative humidity. Raindrop mass evolution estimated from the model shows that, on average, 15 % of rain mass evaporates over Pune.*

Observational measurements are less common and highly valuable, and thus the sampling of rain and vapor isotopes is compelling. The authors have produced an observational dataset of high value, and the measurements are likely to provide a valuable and novel contribution to this research field. The authors can strengthen their findings from the observational data by further substantiating points #1-4 and #6 of the conclusions.

**R-2-5:** We appreciate the comment and would try to follow up on the suggestion. The conclusion part (#1 to #4 and $6) has been elaborated as suggested. Please see below.

1. The vapour isotopes show considerable temporal variation (with $\delta^{18}O$ from -19.2 ‰ to -9.4 ‰ and $\delta D$ from -123.7 ‰ to -63.4 ‰). The corresponding rain isotope variations are (with $\delta^{18}O$ from -7.5 ‰ to 1.2 ‰ and $\delta D$ from -58.9 ‰ to 11.8 ‰). The rain isotopes

are not in equilibrium with the ground vapour. In most cases, the observed rain $\delta^{18}O$ values are higher than expected from equilibrium (by about 2 ‰ on average); in contrast, the $\delta D$ values are not much higher on average (~0.7‰) but there are some extreme values which are higher by about 20 ‰ and lower by about 15 ‰. The deviation from equilibrium is caused by fractionation that raindrops suffer through evaporative exchange (involving equilibrium and kinetic fractionations) during their fall. This fractionation changes the isotope ratios of droplets originated at ~850 mb.

Considering seasonal variations, there were four events extending over a few days when both rain and vapour isotope ratios were considerably lower (for example, rain values were less than the mean - 0.5 standard deviation; with $\delta^{18}O$ < -2.6 ‰). These events seem to be related with regional meteorological characteristics (see below)

2. We note that the low rain isotope events are found to be synchronous with negative OLR anomalies. Negative OLR anomalies during Indian summer monsoon are usually associated with large-scale convections (Sengupta et al., 2020). The intense convective events, indicated by these anomalies, lift the air parcels to higher altitudes where the surrounding vapour isotope ratios are highly depleted. Droplets formed from these vapours are correspondingly depleted. The drop isotope values may be so negative that even evaporative exchanges may not alter/increase their values much on the way down, especially if they are of big size. Both ground vapour and rains display high negative isotope values indicating downdraft associated with such evaporative cooling. An indication of lifting air parcels to higher altitudes is marked by a second CLWC peak (about 550 mb; see Figure R-2-1) for 19, 25, 27 Sept, 2019 when the rain and vapour isotopes were highly depleted (rain $\delta D$ values are -39 ‰, -49 ‰ and -59 ‰ compared to the average of -7‰) and associated with the negative OLR anomaly (Fig R-2-1). It shows that on those days intense convection lifted the moist air parcels to about 5.5 km which produced significant CLWC peaks.

[Figure]

Figure R-2-1: Presence of second CLWC peaks at a high altitude (~500 mb) on 19, 25 and 27 Sept, 2019 when highly depleted rain $\delta$D values are observed in association with negative OLR anomaly.

3. A gradual increase in the d-excess values of vapour and a small but notable decrease in $\delta^{18}$O values in the later part of the monsoon (after mid-August) are observed. The high vapour d-excess in September is especially noticeable. We also find a strong anti-correlation between vapour $\delta^{18}$O and vapour d-excess values. Such anti-correlation usually indicates a significant contribution to the ground vapour from evaporative sources near the last phase of monsoon. Pathak et al. (2014) found a higher precipitation recycling ratio, that is, the ratio of recycled precipitation to total precipitation in central India, at the end of the monsoon (September). In contrast, the rain d-excess values are not significantly different because there is no such cumulative effect for the rains.

4. The above observations suggest increased vapour contribution from evaporation of raindrops and/or from local evapotranspiration (ET) sources, especially increasing with the monsoon progress. However, vapour supply from surface sources cannot be a large factor. Pathak et al. (2014) showed that for central India the ET sources can at best contribute 5% to 10%; the contribution increases with the progress of the Monsoon. Therefore, we strongly believe that downdraft of depleted vapour (and not local supply) is the main source of low $\delta^{18}$O (and high d-excess) surface vapour (Risi et al., 2023).

The depleted vapour in the sub-cloud region can originate from raindrop evaporation and such vapour can be down-drafted by the drag of the falling raindrops.

6. In the Δδ-Δd (Δ indicates the vapour in equilibrium with rain minus the ambient vapour, following the definition of Graf et al., 2019) cross plot, about half of the data points lie in the lower right quadrant, which signifies the dominance of raindrop evaporation over Pune and the adjoining region during our study period. The distribution of points in this quadrant is indicative of drop evaporation but this fact alone cannot quantify the magnitude. The slope of the points (about -0.45) suggests that evaporation is intense. This is because a higher slope in the cross plot is caused by a relatively magnified effect of d-excess difference between the rain (and corresponding equilibrated vapour) and the ambient vapour caused by a larger evaporation. A comparison can be made with the study of Graf et al. (2019) who found a lower slope at a value of -0.3 for Zurich.

The slope is essentially due to a differential effect in evaporative fractionation. Evaporation decreases rain d-excess but increases rain $\delta D$. However, the magnitudes of changes, negative for d-excess and positive for $\delta D$, are not the same. Fractionation values (involving equilibrium and kinetic factors) show that the change in $\delta D$ is larger and that in d-excess is lower (about 30% of the $\delta D$ change, considering only the absolute values for the changes). This is because in evaporation, the kinetic effect operates in addition to the equilibrium fractionation and that has more influence on $\delta D$ compared to $\delta^{18}O$. If the evaporation is higher (due to higher temp and lower RH) the deviation from the equilibrium fractionation line will be more and the slope will be higher. In frontal systems of Switzerland, the temperature was about 12°C and RH about 80% (from Graf et al, 2019) compared to Pune where T was about 25° C and RH about 85%. Since we know that temperature plays an equal role as RH for evaporation (see coefficients of the normalized evaporation equation, 0.370 against -0.329 in the multiple regression equation) for the large increase in Pune temperature we expect more evaporation; this leads to the higher slope value of -0.45 for Pune compared to -0.30 for Zurich. Pune slope calculated by Bootstrap method (see reply R-2-17) shows that the best candidate for the slope is -0.45. The multiple regression equation is:

Evaporation Fraction = -0.329*RH +0.370* Temperature -0.665* Diameter

More care should be put into describing the data collection methods and ascertaining firm conclusions on the observed trends within the context of other literature.

**R-2-6:** The data collection method has been elaborated (please see the replies below related to the specific queries on Research Methodology).

**Research Methodology:**
Below are specific comments on the research methodology, first for the experimental component, and then for the modeling.
Observational measurements:
The measurements in this study that are compelling:
● Rainwater isotope fraction and d-excess: I'm convinced that these measurements are highly reliable, although the authors should clarify if the rainwater is collected every 24 hours, and how this is done (e.g. what time of day is the sample collected?). Also, the link to the IAEA document is not easily accessible (I was not able to access it on a number of different web browsers and devices). So, it would be good to briefly summarize the collection method in case the reader can't access the link.

**R-2-7:** We are sorry that the IAEA link does not work anymore. For clarity, we now elaborate on what we did. The rain samples are collected during the rainy days (on average about 50 rainy days per season in Pune) over 24 hr interval (8:30 am to 8.30 am next day). The collection methods are now given in more detail including two schematic diagrams showing rain and vapors sample collectors. The procedure of rain sample collection is given below.

[Figure]

Fig. R-2-2: Schematic showing the set up for rain sample collection

***Rain sample collection procedure***

*Rain samples were collected on a daily basis at 8:30 A.M (Indian Standard Time) using rain samplers made as per the guidelines of the International Atomic Energy Agency. For clarity, a schematic of the setup is presented in Fig. R-2-2. Samples were collected for all rainy days (rain rate ≥ 0.1 mm/day). A 10-Litre carboy bottle fitted with a 25 cm diameter funnel was fixed ~1 meter above the ground level to avoid contribution from ground splashing. A tube was attached to the tip of the funnel that touched the bottom of the plastic bottle, which helped reduce the exposed surface area and minimise evaporation. The rain samples collected in the sampler are transferred into 8 ml Polylab bottles. The name of the location, date and time of collection were labelled on the sample bottles. The samples were measured immediately after collection at IITM.*

*All rainwater samples are measured in a Laser based Liquid Water Isotope Analyzer (LWIA; Model No: TIWA-45-EP), manufactured by Los Gatos Research (LGR) with a routine analytical precision of 0.1‰ and 1‰ for $\delta^{18}O$ and $\delta^{2}H$ respectively using five laboratory standards. These laboratory standards are periodically calibrated with respect to IAEA primary standards. The measurement procedure followed a standard protocol as discussed in numerous publications and is briefly described below.*

*The LWIA measures both $\delta^{18}O$ and $\delta^{2}H$ of liquid samples simultaneously, with every measurement taking ~ 1.5 minutes. The 1 mL aliquots of all water samples (both rain and standards) are transferred into 2 mL glass vials capped with pre-sealed silicone septa. During measurement, 1.2 µL of liquid is further sampled by an autosampler and injected into an injector block (preheated at ~70°C). Water vapour produced in the injector block is then transferred to the isotope analyser through a Teflon tube. Between two consecutive injections, dry air is passed through the cavity, and the entire cavity is pumped to eliminate the trace of the sample memory from previous measurements. In high throughput mode, every sample is measured nine times by nine separate injections, where the first four injections are discarded to eliminate the memory effect. The data measured in the analyser are then calibrated with respect to Vienna Standard Mean Ocean Water (VSMOW) using post-processing software.*

● Ground based rainfall rate, temperature, humidity from IMD: Although not precisely collocated with the rainwater isotope samples, the daily averages are temporally consistent with the rainwater collection and are likely valid.

**R-2-8:** We agree with the reviewer. The IMD observatory is ~4 km away from the vapour sample location.

● Radiosondes: Although not precisely collocated with the rainwater samples, and the mismatch in temporal resolution is severe (24 hr integration for isotopes, vs averaging of two point measurements for sondes), atmospheric profiles in the same region and date-range as the rainwater isotopes provide extremely valuable atmospheric context.

**R-2-9:** We agree on the two limitations, collocation and time mismatch. But as explained above this was the best option available to us from a location close to the sample

collection. The issue related to the mismatch in the temporal resolution has also been addressed below in the next reply (R-2-10).

I think the authors could be more careful with dealing with the temporal discrepancy. For example, on days with precipitation, it would be wise to verify that radiosondes were launched during the time of precipitation. As an extreme example, if the day's precipitation accumulated between 04-09Z, but the sondes are launched at 00Z and 12Z, the atmospheric soundings for that day might be very misrepresentative and may not even contain clouds. In general, I think these radiosondes could be utilized to a greater extent in the observational analysis.

**R-2-10:** We agree that there are limitations on use of RH and T from radiosonde. On operating days, the radiosondes are usually launched at 00Z and 1200Z in Pune. Due to operational challenges, these measurements are not generally carried out when there is rain. The radiosonde data for Pune are expected to be reasonable for use in the model, if we can at least show that the difference between the two consecutive available measurements is not large. The difference between RH ($\Delta$RH) and temperature ($\Delta$T) measured at 1200Z and 00Z are plotted against height in Fig. R-2-3 for the 29 days which are considered in the BCIM runs. The figure shows that the $\Delta$RH values are within $\pm$ 10% on most of the days (~80% of the total sampling days) and $\Delta$T values are within 2°C. This is excepted as those parameters over the western India do not vary much during Indian Summer Monsoon (Pathak et al, 2013). We need to check how serious these differences are in the context of their use in the model. We have shown through sensitivity analyses and two multiple regression analyses (see replies to Reviewer-1, R-1-1 and R-1-14) that the effects of the daily scale variation on model rain isotope values and evaporation fraction are not significant. We also demonstrate further that these RH and T data from the sonde used in this study is more reliable than the same obtained from any satellite datasets (see the reply R-2-27).

[Figure]

**Fig. R-2-3:** The RH and temperature differences (∆RH, in% and ∆T, in °C) between two radiosonde measurements (1200Z data- 00Z data) of each day versus altitude (in mb). These differences for 29 days show that most of the days the differences are not large and a mean value of each day can be used without compromising the quality of model simulations.

Vapor water isotope collection:

● It's possible that these vapor isotope measurements are highly reliable, but more details need to be provided by the authors to allow the reader to make this assessment. It seems that a detailed description of the vapor collection apparatus has not been published prior, and no literature references were provided for the design. As such, including a picture or a diagram of the apparatus would be helpful.

**R-2-11:** The vapour collection procedure along with schematic diagram of the set up are given below. This is added in the revised manuscript.

*"Vapour collection Procedure*

*An indigenously fabricated glass condenser (Figure R-2-4) was used for the collection. One end of the condenser was connected to a diaphragm pump through a PTFE tube. The ambient air was pulled through this glass condenser by the pump at a flow rate of 800 ml/min. The glass sampler (Trap-1) was immersed in alcohol slurry in a precooled Dewar flask. The temperature of the slurry was maintained at -80℃ by using liquid nitrogen. At this temperature, the atmospheric moisture is condensed into ice leaving all other gases which are pumped out*

*from the sampler. To obtain adequate condensed vapour, the sampling was carried out for 3-4 hours, depending upon the daily specific humidity. One sample is collected per day usually between 9-30 am and 2 pm. It was ensured that the amount of collected sample is at least double the minimum amount required for isotope analysis (i.e., about 1ml). The airflow was controlled at 800 ml/min by a Cole Palmer flow meter (model number MR3A138VVTCP) to ensure laminar flow in the collection line. We added another trap (Trap-2) for checking complete collection. A complete collection by Trap-1 is ensured when there is no ice (water) in Trap-2. After sampling, the condenser was brought to room temperature. The water samples were then pipetted out and stored in 2 ml glass vials. During a period of nearly 4 months (July to October), 48 vapour samples were collected."*

[Figure]

Fig. R-2-4: Schematic showing the set up for vapour sample collection

More importantly, were there any calibrations or validation of the apparatus performed prior to collecting the measurements?

A question I have is about the level of certainty that the flow rate, temperature, and surface area within the chilled flask was sufficient to condense out all of the water vapor from the air sample as it's pumped through? If the water vapor is only partially removed

from the sample (with the remainder flowing out), then significant kinetic fractionation could occur during the process of condensing the water vapor inside the dewar. If this were to be the case, the water condensed in the dewar would adopt an enriched bias, and an elevated d-excess that could also have a dependence on humidity. This may possibly explain some of the d-excess trends in the water vapor that were unaccounted for. Can the authors comment on this?

**R-2-12:** The trap is fabricated and associated apparatus (flow meter, valves, pumps etc.) are assembled as per the technical specifications mentioned in Deshpande et al. (2010) who checked the efficiency of the trap. In addition, we added a second (Trap-2) to check  if the condensation in Trap-1 is complete. We checked this by two ways: (1) no condensate should exist in trap-2 and (2) we calculate the amount of vapour expected to be condensed in Trap-1 using ambient specific humidity and flow rate and compare the derived values with the water collected in Trap-1. A table showing these calculations is presented below (Table R-2-1). The table shows a condensation efficiency of 109±15 % and thus proves a complete collection of the vapour in Trap-1 without loss.

Table R-2-1 Calculation of the fraction of water vapour condensed during vapour sampling on different days, using a flow rate of 800 ml min⁻¹ and an air density of 1.18 kg m⁻³

| Date | Volume of water vapour condensed (ml) | Sampling hour (hr) | Relative humidity (%) | Air temperature (°C) | Saturation vapour pressure at air temperature, $e_s$ (hPa) | Actual vapour pressure, e (hPa) | Mixing ratio (w) | Specific humidity (q) g/kg | Volume of air samples (ml) | Mass of air sampled (g) | Mass of water vapour sampled (g) | Mass of the vapour actually condensed (g) | Fraction of water vapour condensed (%)* |
|---|---|---|---|---|---|---|---|---|---|---|---|---|---|
| 22-07 | 4 | 3.5 | 65 | 30.2 | 42.9 | 27.9 | 0.019 | 18.5 | 144000 | 168 | 3.1 | 4.0 | 129 |
| 24-07 | 6 | 3.5 | 80 | 26.6 | 34.8 | 27.9 | 0.019 | 18.4 | 168000 | 198 | 3.7 | 6.0 | 164 |
| 25-07 | 4 | 3.5 | 84 | 25.9 | 33.4 | 28.1 | 0.019 | 18.6 | 168000 | 198 | 3.7 | 4.0 | 108 |
| 26-07 | 3.5 | 3 | 93 | 25.5 | 32.7 | 30.4 | 0.021 | 20.1 | 144000 | 199 | 3.4 | 3.5 | 102 |
| 30-07 | 4 | 3.5 | 91 | 25.6 | 32.8 | 29.9 | 0.020 | 19.8 | 168000 | 199 | 3.9 | 4.0 | 102 |
| 31-07 | 4 | 3.5 | 85 | 26.0 | 33.6 | 28.6 | 0.019 | 18.9 | 168000 | 198 | 3.8 | 4.0 | 106 |
| 01-08 | 3.5 | 3 | 89 | 26.2 | 34.0 | 30.2 | 0.020 | 20.0 | 144000 | 170 | 3.4 | 3.5 | 103 |
| 02-08 | 4.5 | 3.5 | 92 | 26.2 | 33.9 | 31.2 | 0.021 | 20.7 | 168000 | 198 | 4.1 | 4.5 | 110 |
| 05-08 | 4.5 | 3.5 | 97 | 27.0 | 35.6 | 34.5 | 0.023 | 22.9 | 168000 | 198 | 4.5 | 4.5 | 99 |
| 06-08 | 4 | 3.5 | 90 | 26.7 | 34.9 | 31.5 | 0.021 | 20.9 | 168000 | 198 | 4.1 | 4.0 | 97 |
| 07-08 | 4 | 3.5 | 86 | 26.1 | 33.9 | 29.2 | 0.020 | 19.3 | 168000 | 198 | 3.8 | 4.0 | 104 |
| 08-08 | 3.5 | 3 | 84 | 26.7 | 35.1 | 29.5 | 0.020 | 19.5 | 144000 | 170 | 3.3 | 3.5 | 106 |
| 09-08 | 5 | 3.5 | 85 | 26.5 | 34.6 | 29.4 | 0.020 | 19.5 | 168000 | 198 | 3.9 | 5.0 | 130 |
| 13-08 | 3.5 | 3 | 84 | 27.6 | 36.9 | 31.0 | 0.021 | 20.5 | 144000 | 169 | 3.5 | 3.5 | 101 |
| 14-08 | 4.8 | 3.5 | 81 | 27.3 | 36.2 | 29.3 | 0.020 | 19.4 | 168000 | 198 | 3.8 | 4.8 | 125 |
| 21-08 | 4 | 3.5 | 74 | 26.9 | 35.4 | 26.2 | 0.018 | 17.3 | 168000 | 198 | 3.4 | 4.0 | 117 |
| 22-08 | 4.5 | 3.5 | 79 | 27.5 | 36.8 | 29.1 | 0.020 | 19.3 | 168000 | 197 | 3.8 | 4.5 | 118 |
| 23-08 | 3.5 | 3.5 | 71 | 27.0 | 35.7 | 25.3 | 0.017 | 16.7 | 168000 | 198 | 3.3 | 3.5 | 106 |
| 27-08 | 3.5 | 3 | 78 | 27.2 | 36.1 | 28.2 | 0.019 | 18.7 | 144000 | 169 | 3.2 | 3.5 | 111 |
| 30-08 | 4 | 3.5 | 78 | 26.5 | 34.6 | 27.0 | 0.018 | 17.9 | 168000 | 198 | 3.5 | 4.0 | 113 |
| 03-09 | 3.5 | 3 | 84 | 25.9 | 33.4 | 28.0 | 0.019 | 18.6 | 144000 | 170 | 3.2 | 3.5 | 111 |
| 04-09 | 4 | 3.5 | 88 | 26.1 | 33.8 | 29.7 | 0.020 | 19.7 | 168000 | 198 | 3.9 | 4.0 | 102 |
| 05-09 | 3.5 | 3 | 84 | 26.7 | 35.1 | 29.5 | 0.020 | 19.5 | 144000 | 170 | 3.3 | 3.5 | 106 |
| 06-09 | 4 | 3.5 | 87 | 26.3 | 34.3 | 29.8 | 0.020 | 19.7 | 168000 | 198 | 3.9 | 4.0 | 102 |
| 09-09 | 3.5 | 3 | 93 | 26.5 | 34.7 | 32.2 | 0.022 | 21.4 | 144000 | 170 | 3.6 | 3.5 | 96 |
| 16-09 | 3.5 | 3 | 84 | 26.7 | 35.1 | 29.5 | 0.020 | 19.5 | 144000 | 170 | 3.3 | 3.5 | 106 |
| 18-09 | 4 | 3 | 97 | 27.2 | 36.2 | 35.1 | 0.024 | 23.3 | 144000 | 169 | 3.9 | 4.0 | 101 |
| 19-09 | 4.5 | 3.5 | 86 | 26.7 | 35.1 | 30.2 | 0.020 | 20.0 | 168000 | 198 | 4.0 | 4.5 | 114 |
| 25-09 | 3.4 | 3 | 81 | 29.4 | 41.1 | 33.3 | 0.023 | 22.1 | 144000 | 168 | 3.7 | 3.4 | 91 |

*Mean fraction of water vapour condensed 109(±15) %

● More details on the timing of the vapor sample collection is also needed - Were the samples collected once per day? Or during intensive observation periods?

**R-2-13:** One vapour sample was collected per day usually between 9-30 am and 2 pm. Rain and vapour collections were simultaneously done in as many as 29 days.

The authors explain that the local advective vapor fluxes in this region are dominant, and any vapor from evaporating rain is likely negligible in the vapor isotopic signature. Could the authors assume a well-mixed condition below cloud base (assume that vapor isotope ratio is constant below cloud base)?

**R-2-14:** We would like to clarify here that we did not assume a well-mixed condition below the cloud base. As discussed in the text, there are two ways to provide the input

values of various parameters in BCIM: (A) provide only the surface measured values and the model constructs the profile of each parameter considering a Rayleigh ascent (our Run-1), and (B) available profiles of these parameters can be directly used as input profiles (our Runs 2, 3, and 4). As mentioned by the reviewer, a constant isotope composition and specific humidity occur below the cloud base in Rayleigh ascent, ensured by the built-in algorithm (Please see equation A5 in the appendix A1 of Graf et al., 2019) For other runs a small but notable depletion of δD (~12 ‰) is observed corresponding to a change in specific humidity (between the surface and drop introduction altitude). This is in agreement with the earlier finding of Worden et al. (2007) which demonstrated that, in reality, a Rayleigh condition rarely applies and the vapour δD value being a function of specific humidity changes with altitude. In other words, the relationship between specific humidity (q) and δD does not always follow a Rayleigh distillation pattern (Fig.R-2-5a). Although, our input δD profiles are not based on the q profiles (See reply to Reviewer-1, R-1-5 for the methodology), q, as derived from, radiosonde data, also shows upward decreasing trend (Fig. R-2-5b) suggesting that the sub-cloud layer is not well mixed. This is further supported by the radiosonde temperature values which also decrease upwards.

[Figure]

Fig. R-2-5: (a) Scatter plots of δD versus q in tropical continental areas based on TES data (from Worden et al., 2007). Observations are for clear-sky (optical depth 0.2 and humidity < 50%, red symbols) and cloudy (optical depth 0.3 and humidity > 80%, blue symbols) conditions. The solid black curve shows the isotopic composition of saturated vapour in equilibrium with the ocean water for a range of surface temperatures as

marked in °C along the black line (b) specific humidity profiles for 29 days considered in the current study for BCIM runs. These profiles are obtained from radiosonde data. Black bold line indicates mean profile.

In fact, this is the assumption that is made in the Rayleigh model: See Fig 4C of Graf et al 2019, where ambient vapor isotopes (black line) are constant up until the cloud base. A linear temperature profile from the radiosondes would also be indicative that a well-mixed boundary layer assumption is valid.

**R-2-15:** We agree with the reviewer. A well-mixed sub-cloud layer is an assumption in Rayleigh model where below the cloud base the specific humidity is held constant ($q_k$=$q_{k+1}$). But as shown by Graf in his thesis (2017), Rayleigh profile does not pertain to Zurich in reality. Graf mentions in his thesis (Graf, 2017) that for the control simulation (CTRL) of PAY1 and PAY2 (Sections 5.1.5 and 5.2.5), vertical profiles of T and h are taken from the mean of a local area model.

RH, T and specific humidity profiles from Pune radiosonde data do not suggest a well-mixed layer below cloud. See previous reply, R-2-14. This is the reason we used realistic profiles in Run-2, 3, and 4 (RH and T data obtained from radiosonde). The BCIM gives us this flexibility.

The authors mention that the rainwater is not in equilibrium with the vapor at ground level. If the vapor isotope is constant below cloud base, and a temperature profile is known from radiosondes, the authors could identify (based on temperature) the height where the rainwater would be in equilibrium with the vapor. It would be informative to compare this to the lifting condensation level (aka cloud base height), and potentially some qualitative insights on isotopic exchange could be derived.

**R-2-16:** We appreciate the suggestion. But there are two issues here. The departure from equilibrium is so large that a lower temperature (larger enrichment in rain) could not make this happen. As an example, let us take the data from date 22-07-2019. In this case, the $\delta^{18}O(vap)$ of -14.3 ‰ at a ground temperature of 30°C would give equilibrated water with $\delta^{18}O$ = -14.3+8.9= -5.4‰ (8.9 is the fractionation) but the observed $\delta^{18}O(rain)$ is -0.8 ‰, enriched by 4.6 ‰. Now if we go up by 1 km and get a lower temperature of say 24°C (using a lapse rate of ~6°C per km) the expected value would be -14.3+9.4=-4.9 ‰ (9.4 is the fractionation), which is still much lower than -0.8.

It is clear that this high rain $\delta^{18}O$ value cannot derive from the ground vapour by exchange which seems to be too low. On the contrary, the d-excess value of the rain is too low compared to the vapour. This is what is expected from significant drop evaporation.

The assumption that 'vapor isotope is constant below cloud base' does not hold in reality and the suggested exercise would not provide any accurate estimate of LCL and the suggested comparison would be without any physical basis. Actually, in the revised version, as per the suggestion of the reviewer (See R-2-29), we estimate LCL using radiosonde observations and modified our input RH profiles accordingly (that is, assume RH=100% above LCL).

The interpretation of the slope in Line 402-403 and point #6 of the Conclusions section could be strengthened. Fig 7 of Graf et al 2019 shows different conditions that would lead to a steeper slope in the $\Delta\delta$ -$\Delta$d space. The authors have discussed the effects of RH and temperature on the expected slope: What about the ambient vapor isotope values? For example, if the ambient vapor d-excess in Pune is larger than the ambient vapor d-excess in Switzerland, then one would indeed expect the Pune rainfall data to fall along a steeper slope than that in Switzerland. From the observed data presented, it appears that Pune vapor d-excess values range from 10-30 while the data in Graf et al 2019 is in the 5-20 range, so this could be a potential source of the steeper slope. Likewise, higher vapor $\delta$D values would produce a steeper slope, which again the Pune values appear to be larger than those observed in Switzerland. For completeness, all the different factors that might contribute to a steeper slope should be considered and discussed to strengthen this analysis.

**R-2-17:** We should understand the factors responsible for the slope in the $\Delta\delta$-$\Delta$d plot. The slope is essentially due to the following differential effects in evaporative fractionation. Evaporation decreases rain d-excess but increases its $\delta$D. However, the magnitudes of these changes, negative for d-excess and positive for $\delta$D, are not the same. Fractionation values (involving equilibrium and kinetic factors) show that the change in $\delta$D is larger than that in d-excess (about 30% of the $\delta$D change, considering only the absolute values for the changes). This is because in evaporation, the kinetic effect operates in addition to the equilibrium effects and that has more influence on $\delta$D compared to $\delta^{18}O$. If the kinetic fractionation is higher than normal (due to higher temp

and lower RH) the deviation from the equilibrium fractionation line will be more and the slope (in Δδ-Δd plot) will be higher. In frontal systems of Switzerland, the temperature was about 12°C and RH about 80% (from Graf et al., 2019) compared to Pune where temperature was about 25°C and RH was about 85%. Since we know that temperature plays a nearly equal role as RH for evaporation (see coefficients of the normalized evaporation equation (Reply to the Reviewer-1, **R-1-14**, 0.370 against -0.329 in the multiple regression equation shown below) for the large increase in Pune temperature we expect more evaporation; this leads to the higher slope value of -0.45 for Pune compared to -0.30 for Zurich. Pune slope calculated by the Bootstrap method (as given below in Fig. R-2-6) yields a value of -0.45. The multiple regression equation is:

[Figure]

Evaporation Fraction = -0.329*RH +0.370* Temperature -0.665* Diameter

[Figure]

Fig. R-2-6: Distributions of the slope and intercepts for Pune (in Δδ-Δd plot) along with their associated uncertainties calculated by the Bootstrap method. The bootstrap-estimated mean slope is −0.45, which is adopted here.

[Figure]

Fig. R-2-7: Scatter plot showing $\Delta\delta$–$\Delta$d diagram with the slope of the best fit line using excel program as -0.30. We think this estimate is not correct due to presence of extreme values which have undue influence on the slope value.

Our data in simple excel analysis yields a slope of -0.30. But we believe that a bootstrap analysis provides a better slope due to the presence of some highly deviant points. These points are given appropriate status in Bootstrap analysis and therefore we get a more realistic slope value. The Bootstrap analysis was done by taking 20 random sample points each time and finding the slope from fitting those set of points and repeating this 1000 times. The distribution of the slope values along with the mean, median and standard deviation are given in the above plots.

Line 357: Is the ambient temperature used for the $\Delta\delta$ -$\Delta$d plots obtained from the IMD observatory?

**R-2-18:** The line 357 refers to Fig. 4c. In figure 4, the $\Delta\delta$ $^{18}O_{r\text{-}v}$ and $\Delta$d-excess $_{r\text{-}v}$ refers to the $\delta^{18}O$ and d-excess differences of measured rain and vapour samples collected on the same day. No ambient temperature or humidity data are used. This is not the same difference shown in Fig 6 or 7 where we calculate the expected equilibrium vapour assuming drop temperature (which is lower than the ambient temperature taken from the IMD observatory). This is done because equilibrium involves exchange from the

drop surface and therfore the drop temperature should be used for calculation. We have clarified this in a reply to reviewer-1, see R-1-21.

Line 389: Can the authors comment on if the <5mm rain rate of the data in the lower right quadrant is relatively lower than those in the lower left quadrant?

**R-2-19:** We thank the reviewer for this suggestion. In the revised version, we have added a plot (Fig. R-2-7) which shows more samples with <5mm rain rate fall in lower right quadrant compared to that in the left quadrant; this suggests that drop evaporation is a dominant process in low rainfall events (where smaller drop sizes dominate). Motivated by this suggestion, we further analyzed the Δδ -Δd plot in terms of rain rate (Fig. R-2-8). It seems that the higher rainfall points are always in the top left quadrant. This suggests that the memory of the isotopes is partly retained even after evaporation and exchange because higher rainfall implies larger drop size.

Line 390: The text says nine samples, but the scatter plot shows 11 points in the lower left quadrant.

**R-2-20:** We recounted and found that 14 points fall in lower left quadrant (Fig. R-2-7). Thanks for the correction.

[Figure]

Fig. R-2-8: The Δδ -Δd plot for lower left and lower right quadrant with rain rate <5mm/day and >5mm/day

[Figure]

Fig. R-2-9: The Δ$\delta$ -Δd plot for various rain rates. The size of the sample point circles indicates drop size; their variation is associated with the rain rate. Please note most of the 4[th] quadrant (left) points are of bigger sizes and those in the 3[rd] quadrant (right) are mostly of small sizes.

Line 391-392: The statement on size of raindrops and precipitation intensity - Is this a finding from your own observations? If so, please demonstrate how this was assessed.

**R-2-21:** The lines in our paper are:

*"The crucial driving factors for below-cloud processes seem to be the size of raindrops and the intensity of precipitation. This is primarily because raindrops with larger diameters correspond to increased intensity and have shorter residence times in the atmospheric column. As a result, they experience reduced evaporation while descending toward the ground."*

This is not a finding from our study because we did not measure the drop size based on any separate study. Our drop size estimates are based on rain rate or precipitation intensity and are not independent. We found this conclusion from a paper by Law, Kuok and Trinidad (2021): An Experimental Study on The Correlation of Natural Rainfall Intensities and Raindrop Size Distribution Characteristics.  Law et al. (2021 IOP Conf. Ser.: Mater. Sci. Eng. 1101 012009) stated that: "Based on the findings, it can be statistically interpreted that higher rainfall intensities are always characterized by having closer distances among raindrop particles and thus higher chances for smaller raindrops to collide and to coalesce into larger droplets before falling from the cloud".

We are happy to note that our data is consistent with their conclusion and show this by the plot.

**The BCIM model**

In general, the authors should clarify if the radiosondes are being used for the thermodynamic profile in all of these cases (other than the Rayleigh case), or if the GCM, or satellite profiles are being used.

**R-2-22:** For the thermodynamic profiles (RH and temperature) used in Run-2, Run-3 and Run-4, radiosonde data are used (see Table R-2-2). For Run-1 (Rayleigh case), surface measurement of RH and temperature data are used as input and the model derives the profile using the lapse rates and condition for specific humidity variation as summarized in the Appendix A1 of Graf et al. (2019). In particular, for Run-1 specific humidity remains the same for RH<100% but RH increases due to decrease in temperature.

Perhaps the authors could provide a table for each run, specifying the BCIM inputs and where they came from: droplet size, introduction height, Temperature and humidity profile, Deuterium, and O18. Or just be very clear in the description.

**R-2-23:** Thanks for the suggestion. We have provided a table showing the details of all input parameters for various runs, including their sources. See reply R-1-3 in response to reviewer #1. The same table is given below.

Table R-2-2: Input parameters for various BCIM runs

| Number | BCIM input parameter | Data source for Run-1 | Data source for Run-2 | Data source for Run-3 | Data source for Run-4 |
|--------|---------------------|----------------------|----------------------|----------------------|----------------------|
| 1 | Drop size | Marshal-Palmer equation using rain rate from IMD | Marshal-Palmer equation using rain rate from IMD | Marshal-Palmer equation using rain rate from IMD | Marshal-Palmer equation using rain rate from IMD |
| 2 | RH profile | Rayleigh ascent ~15 % increase per km | Average Radiosonde normalized to ground value | Same as Run-2 | Same as Run-2 |
| 3 | Temperature profile | Rayleigh ascent Lapse rate ~ 5.6°C per km | Average Radiosonde normalized to ground value | Same as Run-2 | Same as Run-2 |
| 4 | $\delta D_{vap}$ profile | Rayleigh ascent ~7 ‰ decrease per km | TES normalized to measured ground value | $\delta D$ values reduced slightly to maintain the shapes similar to Run-2 | Nearly the samew profile as Run-3 |
| 5 | d-excess$_{vap}$ profile | Rayleigh ascent ~0.1 ‰ increase per km | LMDZ $\delta D$ and $\delta^{18}O$ values used to get dex vap normalized to measured ground value | d-exc vap decreased from Run-2 average of 17‰ (average ~7‰). | d-exc vap decreased from Run-2 average of 17‰ (average ~10‰). |
| 6 | Rain drop introduction height | ERA-5 Cloud Liquid Water Content peak | ERA-5 Cloud Liquid Water Content peak | ERA-5 Cloud Liquid Water Content peak | ERA-5 Cloud Liquid Water Content peak |
| 7 | Cloud base Height | LCL | LCL | LCL | LCL |

Showing plots of the atmospheric temperature and RH profiles that are being input into the model would be an enormous help to the reader for understanding/assessing the atmospheric structure being modeled. In addition to the isotope profiles shown in Figure S4.

**R-2-24:** Thanks for the suggestion. The input profiles of RH and T for Run-1 and Run-2 are shown below. They will also be given in the revised Manuscript. For Run-3 and Run-4, we have not changed RH and T profiles (please see Table R-1-1 for input details).

[Figure]

Fig. R-2-10: Input profiles of relative humidity (**a and b**) and temperature (**c and d**) for various days in Run-1 and Run-2. The bold red line indicates the average profiles.

If I understood correctly, run-2 uses radiosonde for the thermodynamic profile, dD comes from satellite data from 2005-2009, and d18O comes from the d-excess of the GCM extrapolated to the observed ground-based vapor measurement. There are major concerns with these inputs coming from different data products that all have different spatial/temporal scales and measurement principles.

**R-2-25:** We understand the concern of the reviewer. In response, we shall justify our choice in the following. When any atmospheric model (Global Circulation Model, Regional Climate Model, or the one-dimensional BCIM) is initialized, by necessity, the input parameters from various sources are used, which may have different spatial and temporal resolutions and measurement principles. Moreover, datasets from various sources are also utilized in the atmospheric model across different parametrization

schemes and during nudging. Nudging is a well-known technique where the model values are nudged to accord with the observed values where available. For example, Graf et al. (2019) used point-based radiosonde RH and T observations, as well as isotope outputs from a limited-area model (Pfahl et al., 2012; Villiger et al., 2023), with a km-scale resolution, as input to BCIM. However, the two datasets have different scales and measurement principles. The BCIM has this flexibility in the simulation for using various kinds of dataset which a GCM or RCM does not have. Graf et al. (2019) have considered three sources of RH and T. In the words of Graf in his thesis (2017): "Taking T and RH profiles from radiosonde measurements are the most accurate of the three approaches, because it measures the profiles directly. The close representation of the reality comes at the cost of a limited temporal resolution. Also, a radiosonde profile is not perfectly vertical and not along the fall trajectory of the sampled hydrometeor. Thus, it does not measure the exact profile that a hydrometeor encountered during its fall. A higher temporal resolution can be achieved by taking surface measurements of T and RH and use them as starting values of a (moist- adiabatically ascending air parcel. This combination between model and measurement allows the representation of short-term changes of T and RH." Guided by their argument, we have taken the Radiosonde profiles of each sampling day as our choice and adjusted the lowermost parts to match the measured RH and T values at the ground taken from the IMD.

For example, if the thermodynamic profile and the dD profile were both taken from satellite retrievals, then that dataset would at least be internally consistent. However, considering the varying atmospheric properties from different sources, there is little physical basis to believe that they are internally related to each other.

**R-2-26:** Following the suggestion of the reviewer, we extracted specific humidity and temperature data from Tropospheric Emission Spectra (TES) over the same 4° x 4° box and for the same time period. Relative humidity using the above parameters is further calculated (as the RH data is not directly available from TES). The area- and time-averaged RH and temperature profiles are then compared with those obtained from the Radiosonde observations (Fig. R-2-11). The figure shows that temperature profiles are different at the most by 1°C and RH profiles on average by 8%. Using the multiple regression equation with normalized variables (see reply R-2-38) this can change the model rain $\delta$D values by <1‰. However, in case the reviewer wants, we can run the

model for all 29 days using these new RH and T mean profiles (as obtained from TES) normalized at ground with their surface measured values. But this exercise will take some time.

[Figure]

Fig. R-2-11 Vertical profiles of averaged values of (a) relative humidity and (b) temperature for radiosonde observations (green triangles) with TES satellite retrievals (orange filled circles).

In my view, there are large sources of error being introduced with each model input assumption, starting with the droplet generation height, continuing with the droplet sizes, and compounding with each additional data product incorporated.

**R-2-27: Concern about the droplet generation height:** We agree that one should use the CLWC peak level as the drop introduction height. This has been suggested by the reviewer himself in the paragraph below. We are now using that level and run our model to check the impact it has on the results of rain isotopes. Please see Table R-2-2 below for further clarification on this.

**Concern about droplet size**: The reviewer has raised some concern about drop sizes. We give detailed answers in R-2-35 where we explain the issues involved. In this context, we want to note that Graf et al. (2019) also used the M-P relation for the drop size. Also please see the paper: Sharma et al. (2025) who show that M-P relation gives the best estimate for the rain intensity for monsoon India which in turn is used to obtain the drop diameter. In the absence of disdrometer, the rain rate and M-P relation offers the best choice.  Again, the error introduced has been evaluated and shown in the revised manuscript.

**Concern about RH, T and isotope profiles:** These have been addressed in detail in R-2-25 and in our response to Reviewer #1 in R-1-5. Please note that we always constrain these profiles with the surface level ground observations as boundary conditions. We do not use them as such from the sources but use only their shapes to derive the profiles. This boundary condition constraint improves the model performance as there is no free parameter.

I feel these uncertainties are too severe to enable a useful comparison with the in-situ rainwater measurements. Likewise, the measurements are unable to offer an advantage in assessing the model performance. For these reasons, this reviewer does not see added value in presenting the BCIM in its current form alongside the in-situ rainwater measurements.

**R-2-28:** We are not sure that the reviewer is right about his feeling that the "uncertainties are too severe". How do we verify his assertion? As a test, we tried out two choices for RH (using earlier RH choice and present modified RH using LCL and CLWC peak) and estimated the differences. We found that the $\delta D$ of rain values differ only by about 2 ‰. Please see the diagram in our response below for a related question (Fig. R-2-13). The reason for the small change is a cancellation effect- the vapour $\delta D$ reduces the drop value whereas the lower RH increases the drop value (through enhanced drop evaporation).

**L 426-427**

Clarification on the choice of droplet formation height: I would agree that raindrops are most likely to be generated in regions where CLWC is maximum, but Section 2.4 indicates that the BCIM releases the droplets from cloud base, and it seems implied that this height is what is input to the model. Is the droplet released from the cloud base, or from a height inside the cloud? Showing the input profiles for temperature and RH, as mentioned above would clarify this for the reader.

**R-2-29:** In BCIM, there is no provision to mark/define cloud base height or lifting condensation level (LCL). The model requires a drop formation altitude. If the RH below the drop formation height is <100%, it is considered that the drops are falling through an unsaturated atmosphere and the model applies the equations associated with raindrop evaporation and kinetic fractionations (as if the drops are falling from the cloud base).

On the other hand, If the RH=100% below the drop formation height (let us say from CLWC peak to the cloud base), the model considers an exchange equilibrium fractionation between the droplets and ambient vapour. The drops change their isotope ratios by exchange with the ambient vapour. The rapidity of exchange induced change in this zone depends on the drop size. Bigger drops may retain the memory, but smaller ones will change quickly. In the earlier submitted MS, we considered the droplets formed at CLWC, but as the RH profiles from radiosonde never reach 100% at any altitude during our sampling days, the CLWC peaks and cloud base heights were assumed to be the same. This assumption introduces some errors.

In the revised version, as per the suggestion of the reviewer, we have used the RH, T profiles from the fitted radiosonde data at various heights (with the ground boundary conditions) and estimated the LCL using Skew T-Log P diagram for all 29 days (where we have both rain and vapour isotope observations). A table is appended below where the LCL (assumed to be the cloud base height) and CLWC peak (assumed to be the drop introduction height) are shown (based on ERA5 data). Now it is very clear that Radiosonde RH is never 100% at the LCL (estimated from the Skew-T Log-P diagram). The LCL is taken to be the level where there is proximity (not necessarily the intersection) between the dew point temperature and ambient temperature. Now the pertinent question for application to BCIM would be what RH we will consider between the drop formation height and LCL and below the LCL? Considering the reference RH profile presented by Graf et al (2009) for Rayleigh model, we can consider RH=100% between the LCL and drop formation height. This assumption allows the model to consider an equilibrium fractionation between the droplets and ambient vapour within this zone which is realistic. Below LCL, we can consider the RH as we obtain from the radiosonde measurements. It is possible that the radiosonde RH may not meet RH=100% at the LCL. There will then be a jump from lower value (<100%) to 100%. Therefore, we have to make a small adjustment in RH profiles (Fig. R-2-10) to follow the suggestion of the reviewer. This was done in Run-5.

Table R-2-3: Lifting Condensation level (LCL) and height at which cloud liquid water content (CLWC) is maximum for 29 days. The LCL's are estimated from the radiosonde analysis and CLWC peak data are obtained from ERA5 gridded dataset.

| No. of Days | Date | CLWC Peak (mbar) | LCL (mbar) |
|---|---|---|---|
| 1 | 22-07-2019 | 800 | 823 |
| 2 | 24-07-2019 | 880 | 884 |
| 3 | 25-07-2019 | 900 | 900 |
| 4 | 26-07-2019 | 900 | 900 |
| 5 | 30-07-2019 | 875 | 900 |
| 6 | 31-07-2019 | 880 | 899 |
| 7 | 01-08-2019 | 875 | 884 |
| 8 | 02-08-2019 | 875 | 900 |
| 9 | 05-08-2019 | 850 | 900 |
| 10 | 06-08-2019 | 875 | 900 |
| 11 | 07-08-2019 | 875 | 878 |
| 12 | 08-08-2019 | 880 | 882 |
| 13 | 09-08-2019 | 875 | 889 |
| 14 | 13-08-2019 | 775 | 877 |
| 15 | 14-08-2019 | 850 | 879 |
| 16 | 21-08-2019 | 875 | 879 |
| 17 | 22-08-2019 | 850 | 900 |
| 18 | 23-08-2019 | 720 | 863 |
| 19 | 27-08-2019 | 700 | 882 |
| 20 | 30-08-2019 | 700 | 900 |
| 21 | 03-09-2019 | 894 | 897 |
| 22 | 04-09-2019 | 900 | 900 |
| 23 | 05-09-2019 | 700 | 876 |
| 24 | 06-09-2019 | 850 | 880 |
| 25 | 09-09-2019 | 700 | 898 |
| 26 | 16-09-2019 | 800 | 882 |
| 27 | 18-09-2019 | 825 | 900 |
| 28 | 19-09-2019 | 825 | 900 |
| 29 | 25-09-2019 | 850 | 880 |

Let us restate what we did following the suggestion. We assumed that RH = 100% above the LCL (or cloud base) up to the CLWC peak (drop introduction height) for each day and used the Radiosonde RH below the LCL. We thus obtained the input RH

profiles for each day, as shown in Fig. R-2-10b (in reply R-2-24). Using the new RH profiles and drop introduction heights and earlier vapour $\delta D$, d-excess and T profiles of Run-2, we ran the BCIM again (Run-5). We compare the Run-2 and Run-5 results in Fig. R-2-12 below. There is no notable difference between Run-2 and Run-5. It was, of course, expected as LCL and CLWC peaks are very close for majority of the days, as the reviewer pointed out. Based on these results, we do not intend to change the discussion of Run-3 and Run-4 at this stage. But if the reviewer suggests that we show the calculations of Run-5 in the revised manuscript, we can present the data as a new Run and discuss the results. As of now, we append the results of Run-5 in response to the suggestion of the reviewer.

Diagram for Run-5 and Run-2 rain $\delta D$ are given as below.

[Figure]

Fig. R-2-12. Scatter plot showing observed and simulated **(a)** rain $\delta^{18}O$, **(b)** rain $\delta D$, **(c)** rain d-excess, and **(d)** same data in $\Delta\delta-\Delta d$ diagram. in which input profiles of T, RH, and vapour $\delta D$ and d-excess values are obtained by adopted TES and LMDZ outputs (see text). Sample plots (e-h) with sample T, $\delta D$, and d-excess values and modified RH values (as per the radiosonde-based LCL and CLWC peaks) and drop introduction heights (as per CLWC peaks).

[Figure]

Fig. R-2-13: Run-5 uses new drop introduction height and its rain $\delta D$ value is slightly higher (< 2‰). Run-2 and Run-5 model predictions are well-correlated.

Q: If the droplet is released from cloud base:

The moist adiabatic lapse rate in the Rayleigh method should provide a cloud base height based on the height where the moist parcel reaches saturation, would it not? Could that be used in place of this fixed 850 mb level?

**R-2-30:** In response to this suggestion, we estimated lifting condensation level (cloud base height) using radiosonde data in the revised version (please see the previous reply, R-2-29). It is more accurate and realistic than that estimated using dry adiabatic lapse rate in the Rayleigh method. Instead of taking a fixed 850 mb level for drop formation height, we consider CLWC peak height and re-ran the model with revised LCLs and drop introduction heights (Table R-2-3).

This issue has been discussed above.

If radiosondes from the study period are provided, why not use the lifting condensation level (LCL) from the soundings to define the cloud base height?

**R-2-31:** We appreciate the reviewer's suggestion. We estimated the LCL now from the radiosonde observations (see Table R-2-3).

If the droplet is released from a height inside the cloud:
The authors acknowledge that assuming the same droplet formation height for each day is going to be erroneous. How does this fixed 850 mb height compare with the cloud base height from the moist adiabatic assumption and the cloud base height from the radiosondes on each day?

**R-2-32:** As per the suggestion of the reviewer we estimated LCL from the radiosonde data and considered the CLWC peak height as the drop formation height. A list of LCLs and CLWC peak heights for 29 days is presented in table R-2-3. This table is used further for modifying RH profiles and drop formation heights to be used for each day. We ran the model with these revised profiles and heights (see reply R-2-29).

Can the authors confirm that it is above the cloud base in all cases? The ERA5 data in Figure S3 indicates that the cloud base is not often above 850 mb, which is encouraging, but it does vary. Perhaps the authors could instead calculate the average height above cloud base where CLWC peaks, allowing the droplet formation height to be adjusted based on the cloud base height on each day.

**R-2-33:** Yes, the drop formation height is always above cloud base heights (LCLs) by 0 to 200 meter (average 54 m) and never below. Please see Table R-2-3 and replies in R-2-29 to R-2-32. We thank the reviewer for this important suggestion which prompted us to explore a new definition of cloud base and drop introduction height and what their altitude difference would imply. We carried out rigorous calculation of LCL based on Radiosonde using Skew T-Log P diagrams (http://weather.uwyo.edu/upperair/sounding.html). CLWC peaks were obtained from ERA5 and the two together are given in Table R-2-3. As pointed out by the reviewer, the CLWC peaks are indeed above LCL by about 54 meter on average (there are six extreme cases where the difference is more than 100 m; there are nine cases where the difference is close to zero). We have followed up on the suggested recalculation (See Figure R-2-13). Surprisingly, the rain $\delta D$ differ by only about 2‰, as mentioned above.

**L 431-437**

A citation is needed for the Marshall-Palmer relationship. Looking at the Marshall & Palmer 1948 paper, it may not be an ideal choice for the droplet diameter estimate. Have the authors considered using droplet size distributions from Murali Krishna et. al. 2021, which is included in the references? It appears that Murali Krishna et. al. 2021 has DSD's for a range of rain rates in the Western Ghats which would seem more appropriate than those collected in Ottawa in the Marshall & Palmer paper, and of higher quality using modern measurement techniques.

**R-2-34:** The reference for the Marshall-Palmer relationship is: Marshall, J. S., and W. M.K. Palmer, 1948: The distribution of raindrops with size. J. Meteor. 5, 165-166.

Murali Krishna et. al. 2021, provided drop size distribution for various rain rates which was obtained from a Joss–Waldvogel disdrometer (JWD) at Mahabaleshwar for 2012-2015. The mean drop size was estimated from the rain rate, assuming a Gamma distribution. Unfortunately, no drop size (disdrometer) measurements were available over Pune during the 2019 monsoon, as mentioned in our manuscript. Our choice of drop size was guided by the fact that in various modelling and observational studies, the Marshall-Palmer distribution has been used in the absence of disdrometer measurements (Graf et al., 2019; Sarkar et al., 2023; Morrison et al., 2020; Ryu et al., 2025; Jiang et al., 2024)

**L 443-448**

Can the authors please describe or reference the equations used to calculate these profiles.

**R-2-35:** The related equations are given in Appendix A1 (Initial Condition) of Graf et al. (2019). We have added this reference in the revised version. Basically, vertical profiles of pressure, temperature, humidity, and the isotopic composition of water vapour are calculated from an adiabatic ascent of an air parcel with a given surface composition. A dry adiabatic ascent is allowed in the initial phase until saturation is reached at the cloud base. Above the base a moist-adiabatic lapse rate is used. The involved parameters are: dry and moist lapse rates calculated based on gravitational constant, specific heat of dry air, mass mixing ratio of vapour, latent heat of evaporation, specific gas constant of dry air, temperature, ratio of specific gas constants of dry air and water vapour, specific humidity and saturation vapour pressure.

**Line 470-471**

The authors should provide a brief description and references for the isotope enabled LMDZ GCM model.

R-2-36: The following discussion is added with a reference-

The LMDZ isotope-enabled general circulation model (GCM), known as **LMDZ-iso (Risi et al., 2010)**, is a version of the LMDZ atmospheric model adapted to simulate the natural variations of water isotopes (like $\delta^{18}O$ and $\delta^2H$) in precipitation and vapor. The dynamical equations are discretized in a latitude-longitude grid, with a standard resolution of 2.5° × 3.75° and 19 vertical levels. Water in its vapor and condensed forms is advected by the Van Leer advection scheme (**Van Leer, 1977**), which is a monotonic second-order finite volume scheme. The physical package is described in detail by **Hourdin et al. (2006)**. It includes in particular the Emanuel convective parameterization (**Emanuel, 1991**; **Grandpeix et al., 2004**) coupled to the **Bony and Emanuel (2001)** cloud scheme. Wind (u, v) are further nudged by ECMWF. Water isotopic species ($H_2^{16}O$, $H_2^{18}O$ and HDO) are transported and mixed passively by the large-scale advection and various air mass fluxes. Apart from rain and vapour isotopes, the GCM outputs are available for various atmospheric parameters such as rainfall, temperature etc.).

Referencing a personal communication to obtain the isotopic inputs to the BCIM is a bit peculiar. If this particular GCM output has not been published prior, and was provided for use and publication in this analysis, this reviewer wonders if it would be more appropriate to include the contributor of the GCM isotope profiles as a co-author.

R-2-37: We are fortunate to point out that such single location GCM output (for a small period) was provided to us free of charge by personal request and the contributors did not want to involve them as co-authors. Being co-author means an involvement which is more than the GCM part and acceptance of all other parts of the manuscript. We, of course, thank them in the Acknowledgement. Similar help was obtained from the Swiss team (Prof. Harald Sodeman) who provided the BCIM codes and its updates. However, following the suggestion, we have added the following reference for the LMDZ dataset:

"*Risi C, Bony S, Vimeux F, Jouzel J (2010) Water-stable isotopes in the LMDZ4 general circulation model: Model evaluation for present day and past climates*

*and applications to climatic interpretations of tropical isotopic records. J Geophys Res-Atmos. https://doi.org/ 10.1029/2009JD013255."*

**Line 522, 556-559**

I find the assertion that "the main source of error in Run 1 and Run 2 could be improper vapor isotope values", to be invalid. Considering the uncertainties around the choice of droplet introduction height, droplet size, and thermodynamic profiles from the radiosondes, there are many possible sources for the lack of agreement with the observations. It is very likely a combination of many sources. The conclusions of L 556-559 are not justified, for these same reasons.

**R-2-38:** We appreciate the comment. We agree that the disagreement is likely a combination of many sources. Actually, this sentence was introduced to justify why we need to tune only the vapour isotope and need not bother about other factors in the first order. This is because vapour influence was ten times more than others as shown below (see the reply to reviewer-1. R-1-1).

We tried to estimate the magnitude of individual contributions from these sources. Since we have many sampling days we can estimate the influence by a multiple regression analysis. The analysis was done by using normalized values from the model parameters obtained by Run-4 and we get the following joint regression equation:

$$\delta D_{mod-rain} = -0.114 \cdot RH + 0.035 \cdot Temperature - 0.059 \cdot diameter + 0.986 \cdot \delta D_v$$

This shows the impact of each of those four parameters (surface values) on the $\delta D$ model rain. For example, if we have a 10% higher vapour isotope value at the ground, the rain $\delta D$ would be higher by 9.9%. Using the sample vapour $\delta D$ value (absolute) of -95.8‰ on 22$^{nd}$ July 2019, this means that if the value was higher by 10%, i.e., -95.8+9.6=-86.2‰ (keeping other values constant) the rain $\delta D$ value would have changed from -3.3 to -3.3+0.3=-3‰. The vapour coefficient is the maximum and therefore its impact is maximum on the rain value. This is what is meant by the statement "the main source of error in Run 1 and Run 2 could be improper vapor isotope values". It does not mean that other parameters (RH, Temp and drop diameter) do not have any impact. It is only that they have less impact.

The impact of the drop introduction height has now been carried out after the suggestion by the reviewer and is discussed above. However, if we change the drop

introduction height we need to extend the other three profiles as well. But, surprisingly, the conclusion is the same. The rain δD change is small about 2‰. See Figure R-2-14. But it is a combined effect of lower vapour δD, lower RH and higher temperature below LCL. So we cannot say that it is only due to drop introduction height.

[Figure]

Figure R-2-14: The sensitivity of various input parameters on model rain δD values. Four parameters are considered: (a) vapour δD, (b) relative humidity, (c) temperature, and (d) drop diameters. A change of ±5 % (for temperature), ±10 % (for RH) and ±20 % (for vapour δD) was made to understand the change in model rain values.

**Figure S4**

The vapor $\delta$D profiles in panel (b) of Figure S4 show indications of cloud base heights where the profile transitions from being constant, to undergoing Rayleigh distillation within the cloud. This further begs the question of whether or not the droplet is intended to be released at cloud base or not. In many of these profiles it appears that the droplet formation height of 850 mb is quite far above the cloud base.

**R-2-39:** We agree with the points made by the reviewer as explained above in response to his suggestion regarding LCL calculation and taking CLWC peak as the drop

introduction height. We also clarified that we have done the rain isotope calculation using BCIM and summarized the results in Run-5. We showed that surprisingly there is only a small change after carrying out these calculations (see Figure R-2-13). The reason for the change being small was also explained.

This also raises questions about the isotopic profiles used in Runs 2-4 that needs to be discussed: Why are the ambient vapor isotopes below the cloud base decreasing with height (and where is the cloud base)? I don't see a physical basis to justify this given that there are strong/dominant surface moisture fluxes, and the expectation of a well-mixed boundary layer.

**R-2-40:** In replies R-2-14, R-2-15, R-2-29 and R-2-39, we have discussed that well mixed boundary layer or Rayleigh ascent do not occur in reality. Worden et al. (2007) and many other subsequent studies have pointed out that vapour $\delta$D values depend upon specific humidity (Fig. R-2-5a). As specific humidity decreases with altitude, even within the boundary layer (Fig. R-2-5b), a concomitant depletion of vapour $\delta$D is expected. Our radiosonde data show that specific humidity changes from ground to the LCL.

About the issue of cloud base, we now take the LCL as the cloud base in agreement with the suggestion of the reviewer. We also take CLWC peak as the drop introduction height. The radiosonde derived RH profile is terminated at the LCL of the day and from the CLWC peak to the LCL the RH is taken as 100%. This introduces a discontinuity at the LCL which cannot be avoided. However, please note that this is merely an issue of formal naming because BCIM only requires drop introduction height and RH, T and isotope profiles from ground to the drop introduction height in actual runs (non-Rayleigh). These are discussed in detail in the thesis of Graf (2017).

**Figure S7**

I'm not seeing where the dexc-low line is in the plot?

**R-2-41:** The effect of vapour d-excess is small and the line is superposed on the reference line. It can be visible upon magnification. Please see below.

[Figure]

Fig. R-2-15: $\Delta\delta$-$\Delta$d diagram summarizing the results of sensitivity experiments using the BCIM. The black line shows the result of the reference setup, and the coloured lines show simulations with various input parameters. Three points denote three sizes: Solid circle (●) 1.3 mm, open triangle (Δ) 1 mm, and open circle (O) 0.6 mm

**Overall organization:**

The manuscript needs significant reorganization for readability and to meet expected standards. Reorganizing the sections could also result in a much more concise paper. The line of logic leading to each conclusion is rather circuitous (not direct). In my opinion the lack of organization makes for a laborious reading experience.

A few opportunities for re-organization are:

● Lines 263 – 275 is more appropriate for the discussion section, rather than results.

● All of Section 4.3 belongs within the methods and results section.

● Lines 667 – 676 belong in the discussion section, not in conclusions.

**R-2-42:** We agree with the reviewer and changes following all the suggestions will be incorporated in the revised manuscript.

**Line by line comments:**

Check with editor: Should "micro-physics" be hyphenated?

**R-2-43:** Checked and corrected

L 78: capitalization of "Estimate"

**R-2-44:** Not required- corrected.

L 81-82: I think the assessment of previous studies being "often innacurate" is too strong. Suggest for example: "However, it remains a challenge to account for all cloud microphysical processes and their associated isotopic fractionations." The sentence could simply be removed as well.

**R-2-45:** Yes, the reviewer is right. We followed his advice and removed the sentence. Thanks for the suggestion.

L 97: "riming" misspelled

**R-2-46:** Thanks. Corrected

L 109: As the referenced study has four years of 1-min disdrometer data, I think the sentence stating "limited to scanty observations" is too strong. Furthermore in the following sentence, I think it is again too strong to claim the current study as "accurate and simpler". I would suggest avoiding these types of qualifier words. For example, the authors could revise to: "The question arises of whether one can determine the raindrop evaporation and its variation using measurements of isotope ratios in rain and vapor".

**R-2-47:** We agree with reviewer's suggestion and incorporate in the revised manuscript.

L 147: The link to this document does not appear to be working. Perhaps the authors can briefly summarize the design and requirements of the rain collection in case the reader is unable to access the document.

**R-2-48:** Yes- the reviewer is right. It stopped working recently. We have now omitted it all together and gave schematic diagrams of the two set-ups (see replies R-2-7 and R-2-11). Thanks for this correction.

L 150: "indigenously" is not the correct word here. Did the authors mean "in-house"?

**R-2-49:** Yes- thanks for the suggestion.

L 152: units "min-1" needs a superscript.

**R-2-50:** Yes. Thanks

L 351: punctuation.

**R-2-51:** Yes corrected. Thanks

L 566: What is "point #4"?

R-2-52: The phrase 'In the context of point #4,' is deleted in the revised manuscript.

References:

1. Bhat G. S. et al.: BOBMEX: The Bay of Bengal Monsoon Experiment, Bulletin of the American Meteorological Society, 82 (10), 2217–2244, 2001

2. Bony, S., and K. A. Emanuel.: A parameterization of the cloudiness associated with cumulus convection; Evaluation using TOGA COARE data, J. Atmos. Sci., 58, 3158–3183, 2001

3. Deshpande, R. D., Maurya, A. S., Kumar, B., Sarkar, A., and Gupta, S. K.: Rain vapor interaction and vapor source identification using stable isotopes from semiarid western India. Journal of Geophysical Research: Atmospheres, 115, 2010

4. Emanuel, K. A.: A scheme for representing cumulus convection in large-scale models, J. Atmos. Sci., 48, 2313–2329, 1991

5. Graf, P.: The effect of below-cloud processes on short-term variations of stable water isotopes in surface precipitation, PhD Thesis, ETH No. 24777, ETH Zurich, https://doi.org/10.3929/ethz-b-000266387, 2017.

6. Graf, P., Wernli, H., Pfahl, S., and Sodemann, H.: A new interpretative framework for below-cloud effects on stable water isotopes in vapour and rain, Atmos. Chem. Phys., 19, 747–765, https://doi.org/10.5194/acp-19-747-2019, 2019

7. Grandpeix, J. Y., V. Phillips, and R. Tailleux.: Improved mixing representation in Emanuel's convection scheme, Q. J. R. Meteorol. Soc., 130, 3207–3222, 2004

8. Hourdin, F., et al.: The LMDZ4 general circulation model: Climate performance and sensitivity to parametrized physics with emphasis on tropical convection, Clim. Dyn., 27, 787–813, 2006

9. Jiang, Y., Yang, L., Li, J., Zeng, Y., Tong, Z., Li, X., and Li, H.: Diurnal Variation Characteristics of Raindrop Size Distribution Observed by a Parsivel[2] Disdrometer in the Ili River Valley, Advances in Meteorology, 2024, 1481661, 1-15, https://doi.org/10 https://doi.org/10.1155/2024/1481661, 2024

10. Law, S L G., Kuok, K K., and Trinidad, S G.: An Experimental Study on the correlation of natural rainfall intensities and raindrop size distribution

characteristics. IOP Conference Series: Materials Science and Engineering, Volume 1101, The 13th International UNIMAS Engineering Conference 2020 (ENCON 2020) 27th - 28th Oct 2020, Kuching, Malaysia. DOI: 10.1088/1757-899X/1101/1/012009

11. Marshall, J. S., and W. M.K. Palmer.: The distribution of raindrops with size. J. Meteor. 5, 165-166, 1948

12. Midhun M, Lekshmy PR, Ramesh R.: Hydrogen and oxygen isotopic compositions of water vapor over the Bay of Bengal during monsoon. Geophys. Res. Lett., 40:6324–6328. Doi:10.1002 /2013GL058181, 2013

13. Morrison, H., van Lier-Walqui, M., Fridlind, A. M., Grabowski, W. W., Harrington, J. Y., Hoose, C., et al.: Confronting the challenge of modeling cloud and precipitation microphysics. Journal of Advances in Modeling Earth Systems, 12,e2019MS001689. https://doi.org/10.1029/2019MS001689, 2020

14. Murali Krishna, U. V., Das, S. K., Sulochana, E. G., Bhowmik, U., Deshpande, S. M., and Pandithurai, G.: Statistical characteristics of raindrop size distribution over the Western Ghats of India: Wet versus dry spells of the Indian summer monsoon. Atmospheric Chemistry and Physics, 21(6):4741–4757, 2021

15. Nimya, S. S., Sengupta, S., Parekh, A., Bhattacharya, S. K., and Pradhan, R.: Region-specific performances of isotope enabled general circulation models for Indian summer monsoon and the factors controlling isotope biases, Clim. Dynam., 59, 3599–3619, https://doi.org/10.1007/s00382-022-06286-1, 2022.

16. Oza, H., Ganguly, A., Padhya, V., and Deshpande, R.: Hydrometeorological processes and evaporation from falling rain in Indian sub-continent: Insights from stable isotopes and meteorological parameters. J. Hydrology, 591, 125601, 2020a

17. Pfahl, S., Wernli, H., and Yoshimura, K.: The isotopic composition of precipitation from a winter storm –a case study with the limited-area model COSMOiso, Atmos. Chem. Phys., 12, 1629–1648, Doi:10.5194/acp-12-1629-2012, 2012

18. Pathak, A., Ghosh, S., and Kumar, P.: Precipitation Recycling in the Indian Subcontinent during Summer Monsoon, J. Hydrometeorol., 15, 2050–2066, https://doi.org/10.1175/JHM-D-13-0172.1, 2014

19. Rahul, P., Ghosh, P., Bhattacharya, S.K., and Yoshimura, K.: Controlling factors of rainwater and water vapor isotopes at Bangalore, India: Constraints from

observations in 2013 Indian monsoon, J. Geophys. Res.-Atmos., 121, https://doi.org/10.1002/2016JD025352, 2016

20. Risi, C., S. Bony, F. Vimeux, and J. Jouzel.:Water-stable isotopes in the LMDZ4 general circulation model: Model evaluation for present-day and past climates and applications to climatic interpretations of tropical isotopic records, J. Geophys. Res., 115, D12118, DOI:10.1029/2009JD013255, 2010

21. Risi, C., Muller, C., Vimeux, F., Blossey, P., Védeau, G., Dufaux, C., and Abramian, S.: What controls the mesoscale variations in water isotopic composition within tropical cyclones and squall lines? Cloud Resolving Model Simulations in Radiative-Convective Equilibrium, J. Adv. Model Earth Syst., 15, e2022MS003331, https://doi.org/10.1029/2022MS003331, 2023

22. Ryu, S., Song, J.J., and Lee, G.: Radar–Rain gauge merging for high-spatiotemporal-resolution rainfall estimation using radial basis function interpolation, Remote sensing, 17, 530, https://doi.org/10.3390/rs17030530, 2025.

23. Sarkar, M., Bailey, A., Blossey, P., de Szoeke, S. P., Noone, D., Quiñones Meléndez, E., Leandro, M. D., and Chuang, P. Y.: Sub-cloud rain evaporation in the North Atlantic winter trade winds derived by pairing isotopic data with a bin-resolved microphysical model, Atmos. Chem. Phys., 23, 12671–12690, https://doi.org/10.5194/acp-23-12671-2023, 2023

24. Sharma, J., Rastogi, A., Verma, S., Kumar, G., and Choudhary, A.: Assessing the accuracy of different Z-R relationships for Doppler Weather Radar based rainfall estimation: A comparative study for the Delhi region, Physics and Chemistry of the Earth 141 104182, 2025

25. Sengupta, S., Bhattacharya, S. K., Parekh, A., Nimya, S. S., Yoshimura, K., and Sarkar, A.: Signatures of monsoon intra-seasonal oscillation and stratiform process in rain isotope variability in northern Bay of Bengal and their simulation by isotope enabled general circulation model, Clim. Dynam., 55, 1649–1663, https://doi.org/10.1007/s00382-020-05344-w, 2020

26. Sengupta, S., Bhattacharya, S. K., Sunil, N. S., and Sonar, S.: Quantifying Raindrop Evaporation Deficit in General Circulation Models from Observed and Model Rain Isotope Ratios on the West Coast of India, Atmosphere, 14, 1147, https://doi.org/10.3390/atmos14071147, 2023

27. Van Leer, B.: Towards the ultimate conservative difference scheme: IV. A new approach to numerical convection, J. Comput. Phys., 23,276–299, 1977

28. Villiger, L., Dütsch, M., Bony, S et al.: Water isotopic characterisation of the cloud–circulation coupling in the North Atlantic trades – Part 1: A process-oriented evaluation of COSMOiso simulations with EUREC4A observations, Atmos. Chem. Phys., 23, 14643–14672, https://doi.org/10.5194/acp-23-14643-2023, 2023

29. Worden, J., D. Noone, K. Bowman, and R.: Beer Importance of rain evaporation and continental convection in the tropical water cycle, Nature, 445, 528–532,